# FIT LIKE YOU SAMPLE: SAMPLE-EFFICIENT SCORE MATCHING FROM FAST MIXING DIFFUSIONS

## ABSTRACT

Score matching is an approach to learning probability distributions parametrized up to a constant of proportionality (e.g. Energy-Based Models). The idea is to fit the score of the distribution (i.e. $\nabla_x \log p(x)$), rather than the likelihood, thus avoiding the need to evaluate the constant of proportionality. While there's a clear algorithmic benefit, the statistical cost can be steep: recent work by Koehler et al. (2022) showed that for distributions that have poor isoperimetric properties (a large Poincaré or log-Sobolev constant), score matching is substantially statistically less efficient than maximum likelihood. However, many natural realistic distributions, e.g. multimodal distributions as simple as a mixture of two Gaussians in one dimension—have a poor Poincaré constant.

In this paper, we show a close connection between the mixing time of a *broad class of* Markov processes with generator $\mathcal{L}$ and stationary distribution $p$, and an appropriately chosen *generalized score matching loss* that tries to fit $\frac{\mathcal{O}p}{p}$. In the special case of $\mathcal{O} = \nabla_x$, and $\mathcal{L}$ being the generator of Langevin diffusion, this generalizes and recovers the results from Koehler et al. (2022). This allows us to adapt techniques to speed up Markov chains to construct better score-matching losses. In particular, *"preconditioning"* the diffusion can be translated to an appropriate *"preconditioning"* of the score loss. Lifting the chain by adding a temperature like in simulated tempering can be shown to result in a Gaussian-convolution annealed score matching loss, similar to Song & Ermon (2019). Moreover, we show that if the distribution being learned is a finite mixture of Gaussians in $d$ dimensions with a shared covariance, the sample complexity of annealed score matching is polynomial in the ambient dimension, the diameter of the means, and the smallest and largest eigenvalues of the covariance—obviating the Poincaré constant-based lower bounds of the basic score matching loss shown in Koehler et al. (2022).

## 1 INTRODUCTION

Energy-based models (EBMs) are parametric families of probability distributions parametrized up to a constant of proportionality, namely $p_\theta(x) \propto \exp(E_\theta(x))$ for some energy function $E_\theta(x)$. Fitting $\theta$ from data by using the standard approach of maximizing the likelihood of the training data with a gradient-based method requires evaluating $\nabla_\theta \log Z_\theta = \mathbb{E}_{p_\theta}[\nabla_\theta E_\theta(x)]$ — which cannot be done in closed form, and instead Markov Chain Monte Carlo methods are used. Score matching (Hyvärinen, 2005) obviates the need to estimate a partition function, by instead fitting the score of the distribution $\nabla_x \log p(x)$. While there is algorithmic gain, the statistical cost can be substantial. In recent work, Koehler et al. (2022) show that score matching is statistically much less efficient (i.e. the estimation error, given the same number of samples is much bigger) than maximum likelihood when the distribution being estimated has poor isoperimetric properties (i.e. a large Poincaré constant). However, even very simple multimodal distributions like a mixture of two Gaussians with far away means—have a very large Poincaré constant. As many distributions of interest (e.g. images) are multimodal in nature, the score matching estimator is likely to be statistically untenable.

The seminal paper by Song & Ermon (2019) proposes a way to deal with multimodality and manifold structure in the data by annealing: namely, estimating the scores of convolutions of the data distribution with different levels of Gaussian noise. The intuitive explanation they propose is that the distribution smoothed with more Gaussian noise is easier to estimate (as there are no parts of

the distribution that have low coverage by the training data), which should help estimate the score at lower levels of Gaussian noise. However, making this either quantitative or formal seems very challenging. Moreover, Song & Ermon (2019) propose annealing as a fix to another issue: using the score to sample from the distribution using Langevin dynamics is also problematic, as Langevin mixes slowly in the presence of multimodality and low-dimensional manifold structure.

In this paper, we show that there is a deep connection between the *mixing time* of a broad class of continuous, time-homogeneous Markov processes with stationary distribution $p$ and generator $\mathcal{L}$, and the *statistical efficiency* of an appropriately chosen generalized score matching loss (Lyu, 2012) that tries to match $\frac{\mathcal{O}p}{p}$. In the case that $\mathcal{L}$ is the generator of Langevin diffusion, and $\mathcal{O} = \nabla_x$, we recover the results of Koehler et al. (2022). This "dictionary" allows us to design score losses with better statistical behavior, by adapting techniques for speeding up Markov chain convergence — e.g. preconditioning a diffusion and lifting the chain by introducing additional variables. In summary, our contributions are as follows:

1. **A general framework** for designing generalized score matching losses with good sample complexity from fast-mixing diffusions. Precisely, for a broad class of diffusions with generator $\mathcal{L}$ and Poincaré constant $C_P$, we can choose a linear operator $\mathcal{O}$, such that the generalized score matching loss $\frac{1}{2}\mathbb{E}_p \left\| \frac{\mathcal{O}p}{p} - \frac{\mathcal{O}p_\theta}{p_\theta} \right\|_2^2$ has statistical complexity that is a factor $C_P^2$ worse than that of maximum likelihood. (Recall, $C_P$ characterizes the mixing time of the Markov process with generator $\mathcal{L}$ in chi-squared distance.) In particular, for diffusions that look like "preconditioned" Langevin, this results in "appropriately preconditioned" score loss.

2. We analyze a **lifted** diffusion, which introduces a new variable for temperature and provably show **statistical benefits of annealing for score matching**. Precisely, we exhibit continuously-tempered Langevin, a Markov process which mixes in time $\text{poly}(D, d, 1/\lambda_{\min}, \lambda_{\max})$ for finite mixtures of Gaussians in ambient dimension $d$ with identical covariances whose smallest and largest eigenvalues are lower and upper bounded by $\lambda_{\min}$ and $\lambda_{\max}$ respectively, and means lying in a ball of radius $D$. (Note, the bound has no dependence on the number of components.) Moreover, the corresponding generalized score matching loss is a form of annealed score matching loss (Song & Ermon, 2019; Song et al., 2020), with a particular choice of weighing for the different amounts of Gaussian convolution. This is the first result formally showing the statistical benefits of annealing for score matching.

## 1.1 DISCUSSION AND RELATED WORK

Our work draws on and brings together, theoretical developments in understanding score matching, as well as designing and analyzing faster-mixing Markov chains based on strategies in annealing. We discuss the most related lines of work here, and discuss a broader range of related papers in Appendix I.

**Score matching:** Score matching was originally proposed by Hyvärinen (2005), who also provided some conditions under which the estimator is consistent and asymptotically normal. Asymptotic normality is also proven for various kernelized variants of score matching in Barp et al. (2019). Recent work by Koehler et al. (2022) proves that when the family of distributions being fit is rich enough, the statistical sample complexity of score matching is comparable to the sample complexity of maximum likelihood *only* when the distribution satisfies a Poincaré inequality. In particular, even simple bimodal distributions in 1 dimension (like a mixture of 2 Gaussians) can significantly worsen the sample complexity of score matching (*exponential* with respect to mode separation). For restricted parametric families (e.g. exponential families with sufficient statistics consisting of bounded-degree polynomials), recent work (Pabbaraju et al., 2023) showed that score matching can be comparably efficient to maximum likelihood, by leveraging the fact that a restricted version of the Poincaré inequality suffices for good sample complexity.

Theoretical understanding of annealed versions of score matching is still very impoverished. A recent line of work (Lee et al., 2022; 2023; Chen et al., 2022) explores how accurately one can sample using a learned (annealed) score, *if the (population) score loss is successfully minimized*. This line of work can be viewed as a kind of "error propagation" analysis: namely, how much larger the sampling error with a score learned up to some tolerance. It does not provide insight on when the score can be efficiently learned, either in terms of sample complexity or computational complexity.

**Sampling by annealing:** There are a plethora of methods proposed in the literature that use temperature heuristics (Marinari & Parisi, 1992; Neal, 1996; Earl & Deem, 2005) to alleviate the slow mixing of various Markov Chains in the presence of multimodal structure or data lying close to a low-dimensional manifold. A precise understanding of when such strategies have provable benefits, however, is fairly nascent. Most related to our work, in Ge et al. (2018); Lee et al. (2018), the authors show that when a distribution is (close to) a mixture of $K$ Gaussians with identical covariances, the classical simulated tempering chain (Marinari & Parisi, 1992) with temperature annealing (i.e. scaling the log-pdf of the distribution), along with Metropolis-Hastings to swap the temperature in the chain mixes in time poly($K$).

## 2 PRELIMINARIES

### 2.1 GENERALIZED SCORE MATCHING

The conventional score-matching objective (Hyvärinen, 2005) is defined as

$$D_{SM}(p,q) = \frac{1}{2}\mathbb{E}_p \|\nabla_x \log p - \nabla_x \log q\|_2^2 = \frac{1}{2}\mathbb{E}_p \left\| \frac{\nabla_x p}{p} - \frac{\nabla_x q}{q} \right\|_2^2 \tag{1}$$

Note, in this notation, the expression is asymmetric: $p$ is the data distribution, $q$ is the distribution that is being fit. Written like this, it is not clear how to minimize this loss, when we only have access to data samples from $p$. The main observation of Hyvärinen (2005) is that the objective can be rewritten (using integration by parts) in a form that is easy to fit given samples:

$$D_{SM}(p,q) = \mathbb{E}_{X\sim p}\left[ \mathrm{Tr}\,\nabla_x^2 \log q + \frac{1}{2}\|\nabla_x \log q\|^2 \right] + K_p \tag{2}$$

where $K_p$ is some constant independent of $q$. To turn this into an algorithm given samples, one simply solves $\min_{q\in\mathcal{Q}} \mathbb{E}_{X\sim\hat{p}}\left[ \mathrm{Tr}\,\nabla_x^2 \log q + \frac{1}{2}\|\nabla_x \log q\|^2 \right]$ for some parametrized family of distributions $\mathcal{Q}$, where $\hat{p}$ denotes the uniform distribution over the samples from $p$. This objective can be calculated efficiently given samples from $p$, so long as the gradient and Hessian of the log-pdf of $q$ can be efficiently calculated.[1]

Generalized Score Matching, first introduced in Lyu (2012), generalizes $\nabla_x$ to an arbitrary linear operator $\mathcal{O}$:

**Definition 1.** *Let $\mathcal{F}^1$ and $\mathcal{F}^m$ be the space of all scalar-valued and m-variate functions of $x \in \mathbb{R}^d$, respectively. The Generalized Score Matching (GSM) loss with a general linear operator $\mathcal{O} : \mathcal{F}^1 \to \mathcal{F}^m$ is defined as*

$$D_{GSM}(p,q) = \frac{1}{2}\mathbb{E}_p \left\| \frac{\mathcal{O}p}{p} - \frac{\mathcal{O}q}{q} \right\|_2^2 \tag{3}$$

In this paper, we will be considering operators $\mathcal{O}$, such that $(\mathcal{O}g)(x) = B(x)\nabla g(x)$. In other words, the generalized score matching loss will have the form:

$$D_{GSM}(p,q) = \frac{1}{2}\mathbb{E}_p \|B(x)(\nabla_x \log p - \nabla_x \log q)\|_2^2 \tag{4}$$

This can intuitively be thought of as a "preconditioned" version of the score matching loss, notably with a preconditioner function $B(x)$ that is allowed to change at every point $x$. The generalized score matching loss can also be turned into an expression that doesn't require evaluating the pdf of the data distribution (or gradients thereof), using a similar "integration-by-parts" identity:

**Lemma 1** (Integration by parts, Lyu (2012)). *The GSM loss satisfies*

$$D_{GSM}(p,q) = \frac{1}{2}\mathbb{E}_p\left[ \left\|\frac{\mathcal{O}q}{q}\right\|_2^2 - 2\mathcal{O}^+\left(\frac{\mathcal{O}q}{q}\right) \right] + K_p \tag{5}$$

*where $\mathcal{O}^+$ is the adjoint of $\mathcal{O}$ defined by $\langle \mathcal{O}f, g\rangle_{L^2} = \langle f, \mathcal{O}^+g\rangle_{L^2}$.*

---

[1]In many score-based modeling approaches, e.g. (Song & Ermon, 2019; Song et al., 2020) one directly parametrizes the score $\nabla \log q$ instead of the distribution $q$.

Again, for the special case of the family of operators $\mathcal{O}$ in (4), the integration by parts form of the objective can be easily written down explicitly (the proof is provided in Appendix B):

**Lemma 2** (Integration by parts for the GSM in (4)). *The generalized score matching objective in* (4) *satisfies the equality*

$$D_{GSM}(p,q) = \frac{1}{2}\left[\mathbb{E}_p\|B(x)\nabla_x \log q(x)\|^2 + 2\mathbb{E}_p div\left(B(x)^2\nabla_x \log q(x)\right)\right] + K_p$$

## 2.2 CONTINUOUS-TIME MARKOV PROCESSES

In this section, we introduce the key definitions related to continuous-time Markov chains and diffusion processes:

**Definition 2** (Markov semigroup). *We say that a family of functions* $\{P_t(x,y)\}_{t\geq 0}$ *on a state space* $\Omega$ *is a Markov semigroup if* $P_t(x,\cdot)$ *is a distribution on* $\Omega$ *and* $P_{t+s}(x,dy) = \int_\Omega P_t(x,dz)P_s(z,dy)$. *for all* $x, y \in \Omega$ *and* $s, t \geq 0$.

**Definition 3** (Time-homogeneous Markov processes). *A time-homogeneous Markov process* $(X_t)_{t\geq 0}$ *on state space* $\Omega$ *is defined by a Markov semigroup* $\{P_t(x,y)\}_{t\geq 0}$ *as follows: for any measurable* $A \subseteq \Omega$, *we have* $\Pr(X_{s+t} \in A|X_s = x) = P_t(x,A) = \int_A P_t(x,dy)$. *Moreover,* $P_t$ *can be thought of as acting on a function* $g$ *as* $(P_tg)(x) = \mathbb{E}_{P_t(x,\cdot)}[g(y)] = \int_\Omega g(y)P_t(x,dy)$. *Finally, we say that* $p(x)$ *is a stationary distribution if* $X_0 \sim p$ *implies that* $X_t \sim p$ *for all* $t$.

A particularly important class of time-homogeneous Markov processes is given by Itô diffusions, namely stochastic differential equations of the form

$$dX_t = b(X_t)dt + \sigma(X_t)dB_t \tag{6}$$

for a *drift* function $b$, and a *diffusion coefficient* function. In fact, a classical result due to Dynkin (Rogers & Williams (2000), Theorem 13.3) states that **any** "sufficiently regular" time-homogeneous Markov process (specifically, a process whose semigroup is Feller-Dynkin) can be written in the above form. We will be interested in Itô diffusions, whose stationary distribution is a given distribution $p(x) \propto \exp(-f(x))$. Perhaps the most well-known example of such a diffusion is Langevin diffusion:

**Definition 4** (Langevin diffusion). *Langevin diffusion is the stochastic process* $dX_t = -\nabla f(X_t)dt + \sqrt{2}dB_t$, *where* $f : \mathbb{R}^d \to \mathbb{R}$, $dB_t$ *is Brownian motion in* $\mathbb{R}^d$ *with covariance matrix* $I$. *Under mild regularity conditions on* $f$, *the stationary distribution of this process is* $p(x) : \mathbb{R}^N \to \mathbb{R}$, *s.t.* $p(x) \propto e^{-f(x)}$.

In fact, a completeness result due to Ma et al. (2015) states that we can characterize **all** Itô diffusions whose stationary distribution is $p(x) \propto \exp(-f(x))$. Precisely, they show:

**Theorem 1** (Itô diffusions with a given stationary distribution, Ma et al. (2015)). *Any Itô diffusion with stationary distribution* $p(x) \propto \exp(-f(x))$ *can be written in the form:*

$$dX_t = (-(D(X_t) + Q(X_t))\nabla f(X_t) + \Gamma(X_t))\,dt + \sqrt{2D(X_t)}dB_t \tag{7}$$

*where* $\forall x \in \mathbb{R}^d, D(x) \in \mathbb{R}^{d\times d}$ *is a positive-definite matrix,* $\forall x \in \mathbb{R}^d, Q(x)$ *is a skew-symmetric matrix,* $D, Q$ *are differentiable, and* $\Gamma_i(x) := \sum_j \partial_j(D_{ij}(x) + Q_{ij}(x))$.

Intuitively, $D(x)$ can be viewed as "reshaping" the diffusion, whereas $Q$ and $\Gamma$ are "correction terms" to the drift so that the stationary distribution is preserved.

## 2.3 DIRICHLET FORMS AND POINCARÉ INEQUALITIES

**Definition 5.** *The generator* $\mathcal{L}$ *corresponding to Markov semigroup is* $\mathcal{L}g = \lim_{t\to 0}\frac{P_tg-g}{t}$. *Moreover, if* $p$ *is the unique stationary distribution, the Dirichlet form and the variance are respectively*

$$\mathcal{E}(g,h) = -\mathbb{E}_p\langle g, \mathcal{L}h\rangle \text{ and } \text{Var}_p(g) = \mathbb{E}_p(g - \mathbb{E}_pg)^2$$

*We will use the shorthand* $\mathcal{E}(g) := \mathcal{E}(g,g)$.

By Itô's Lemma, the generator of diffusions of the form (7) have the form:

$$(\mathcal{L}g)(x) = \langle -[D(x) + Q(x)]\nabla f(x) + \Gamma(x), \nabla g(x)\rangle + \text{Tr}(D(x)\nabla^2 g(x)) \tag{8}$$

The Dirichlet form for diffusions of the form (7) also has a very convenient form:

**Lemma 3** (Dirichlet form of continuous Markov Process). *An Itô diffusion of the form* (7) *has a Dirichlet form $\mathcal{E}(g) = \mathbb{E}_p \|\sqrt{D(x)} \nabla g(x)\|_2^2$. Notably, for Langevin diffusion, the Dirichlet form is just $\mathcal{E}(g) = \mathbb{E}_p \|\nabla g\|_2^2$.*

For a general diffusion of the form (7), we can think of $D(x)$ as a (point-specific) preconditioner, specifying the norm with respect to which to measure $\nabla g$. The proof of this lemma is given in Appendix B. Versions of the SDEs we consider have appeared in the literature under various names, e.g., Riemannian Langevin (Girolami & Calderhead, 2011) and preconditioned Langevin (Hairer et al., 2007; Beskos et al., 2008), Fisher-adaptive Langevin (Titsias, 2023).

Finally, we define the Poincaré constant, which captures the mixing time of the process in the $\chi^2$-sense:

**Definition 6** (Poincaré inequality). *A continuous-time Markov process satisfies a Poincaré inequality with constant $C$ if for all functions $g$ such that $\mathcal{E}(g)$ is defined (finite), we have $\mathcal{E}(g) \geq \frac{1}{C} \mathrm{Var}_p(g)$. We will abuse notation, and for a Markov process with stationary distribution $p$, denote by $C_P$ the Poincaré constant of $p$, the smallest $C$ such that above Poincaré inequality is satisfied.*

The Poincaré inequality implies exponential ergodicity for the $\chi^2$-divergence, namely: $\chi^2(p_t, p) \leq e^{-2t/C_P} \chi^2(p_0, p)$ where $p$ is the stationary distribution of the chain and $p_t$ is the distribution after running the Markov process for time $t$, starting at $p_0$. We will analyze the Poincaré constant using a decomposition technique similar to the ones employed in Ge et al. (2018); Moitra & Risteski (2020). Intuitively, these results "decompose" the Markov chain by partitioning the state space into sets, such that: (1) the mixing time of the Markov chain inside the sets is good; (2) the "projected" chain, which transitions between sets with probability equal to the probability flow between sets, also mixes fast. An example of such a result is Theorem 6.1 from Ge et al. (2018), reiterated as Theorem 6 in Appendix A.

## 3 A FRAMEWORK FOR ANALYZING GENERALIZED SCORE MATCHING

The goal of this section is to provide a general framework that provides a bound on the sample complexity of a generalized score matching objective with operator $\mathcal{O}$, under the assumption that some Markov process with generator $\mathcal{L}$ mixes fast. For this section, $n$ will denote the number of samples, and $\hat{\mathbb{E}}$ will denote an empirical average, that is the expectation over the $n$ training samples. We will show:

**Theorem 2** (Main, sample complexity bound). *Consider an Itô diffusion of the form* (7) *with stationary distribution $p(x) \propto \exp(-f(x))$ and Poincaré constant $C_P$ with respect to the generator of the Itô diffusion. Consider the generalized score matching loss with operator $(\mathcal{O}g)(x) := \sqrt{D(x)} \nabla g(x)$, namely $D_{GSM}(p, q) = \frac{1}{2} \mathbb{E}_p \left\| \sqrt{D(x)} (\nabla_x \log p - \nabla_x \log q) \right\|_2^2$. Suppose we are optimizing this loss over a parametric family $\{p_\theta : \theta \in \Theta\}$ satisfying:*

1. *(Asymptotic normality) Let $\Theta^*$ be the set of global minima of the generalized score matching loss $D_{GSM}$, that is $\Theta^* = \{\theta^* : D_{GSM}(p, p_{\theta^*}) = \min_{\theta \in \Theta} D_{GSM}(p, p_\theta)\}$. Suppose the generalized score matching loss is asymptotically normal: namely, for every $\theta^* \in \Theta^*$, and every sufficiently small neighborhood $S$ of $\theta^*$, there exists a sufficiently large $n$, such that there is a unique minimizer $\hat{\theta}_n$ of $\hat{\mathbb{E}} l_\theta(x)$ in S, where*

$$l_\theta(x) := \frac{1}{2} \left\| \frac{\mathcal{O} p_\theta(x)}{p_\theta(x)} \right\|_2^2 - 2\mathcal{O}^+ \left( \frac{\mathcal{O} p_\theta(x)}{p_\theta(x)} \right) = \frac{1}{2} \left[ \|\sqrt{D(x)} \nabla_x \log p_\theta(x)\|^2 + 2 div \left( D(x) \nabla_x \log p_\theta(x) \right) \right]$$

   *Furthermore, assume $\hat{\theta}_n$ satisfies $\sqrt{n}(\hat{\theta}_n - \theta^*) \xrightarrow{d} \mathcal{N}(0, \Gamma_{SM})$.*

2. *(Realizibility) At any $\theta^* \in \Theta^*$, we have $p_{\theta^*} = p$.*

*Then, we have[2]:*

$$\|\Gamma_{SM}\|_{OP} \leq 2C_P^2 \|\Gamma_{MLE}\|_{OP}^2 [\|cov(\nabla_\theta \nabla_x \log p_\theta(x) D(x) \nabla_x \log p_\theta(x))\|_{OP}$$
$$+ \|cov(\nabla_\theta \nabla_x \log p_\theta(x)^\top div(D(x)))\|_{OP} + \|cov(\nabla_\theta \mathrm{Tr}[D(x) \nabla_x^2 \log p_\theta(x)])\|_{OP}$$

---

[2]The notation $\mathrm{div} D(x)$ denotes the vector field $\mathbb{R}^d \to \mathbb{R}^d$, s.t. $\mathrm{div} D(x)_i = \sum_j \partial_j D_{ji}(x)$

**Remark 1.** *The two terms on the right hand sides qualitatively capture two intuitive properties necessary for a good sample complexity: the factor involving the covariances can be thought of as a smoothness term capturing how regular the score is as we change the parameters in the family we are fitting; the $C_P$ term captures how the error compounds as we "extrapolate" the score into a probability density function.*

**Remark 2.** *This theorem generalizes Theorem 2 in Koehler et al. (2022), who show the above only in the case of $\mathcal{L}$ being the generator of Langevin (Definition 4), and $\mathcal{O} = \nabla_x$, i.e. when $D_{GSM}$ is the standard score matching loss. Furthermore, they only consider the case of $p_\theta$ being an exponential family, i.e. $p_\theta(x) \propto \exp(\langle \theta, T(x) \rangle)$ for some sufficient statistics $T(x)$.*

**Remark 3.** *Note that if we know $\sqrt{n}(\hat{\theta}_n - \theta^*) \xrightarrow{d} \mathcal{N}(0, \Gamma_{SM})$, we can extract bounds on the expected $\ell_2^2$ distance between $\hat{\theta}_n$ and $\theta^*$. Namely, from Markov's inequality (see e.g., Remark 4 in Koehler et al. (2022)), we have for sufficiently large $n$, with probability at least $0.99$ it holds that $\|\hat{\theta}_n - \theta^*\|_2^2 \leq \frac{\text{Tr}(\Gamma_{SM})}{n}$.*

Some conditions for asymptotic normality can be readily obtained by applying standard results from asymptotic statistics (e.g. Van der Vaart (2000), Theorem 5.23, reiterated as Lemma 5 for completeness).From that lemma, when an estimator $\hat{\theta} = \arg\min \hat{\mathbb{E}} l_\theta(x)$ is asymptotically normal, we have $\sqrt{n}(\hat{\theta} - \theta^*) \xrightarrow{d} \mathcal{N}\left(0, (\nabla_\theta^2 L(\theta^*))^{-1} \text{Cov}(\nabla_\theta \ell(x; \theta^*))(\nabla_\theta^2 L(\theta^*))^{-1}\right)$, where $L(\theta) = \mathbb{E}_\theta l(x)$. Therefore, to bound the spectral norm of $\Gamma_{SM}$, we need to bound the Hessian and covariance terms in the expression above. The latter is a fairly straightforward calculation, and is included in Appendix C. The bound on the Hessian is where the connection to the Poincaré constant manifests:

**Lemma 4** (Bounding Hessian). *The loss $D_{GSM}$ defined in Theorem 2 satisfies*

$$\left[ \nabla_\theta^2 D_{GSM}(p, p_{\theta^*}) \right]^{-1} \preceq C_P \Gamma_{MLE}.$$

*Proof.* To reduce notational clutter, we will drop $_{|\theta = \theta^*}$ since all the functions of $\theta$ are evaluated at $\theta^*$. Consider an arbitrary direction $w$. We have:

$$\langle w, \nabla_\theta^2 D_{GSM}(p, p_\theta) w \rangle \overset{\text{①}}{=} \mathbb{E}_p \| \sqrt{D(x)} \nabla_x \nabla_\theta \log p_\theta(x) w \|_2^2$$
$$\overset{\text{②}}{\geq} \frac{1}{C_P} \text{Var}_p(\langle w, \nabla_\theta \log p_\theta(x) \rangle) \overset{\text{③}}{=} \frac{1}{C_P} w^T \Gamma_{MLE}^{-1} w$$

① follows from a straightforward calculation (in Lemma 13), ② follows from the definition of Poincaré inequality of a diffusion process with Dirichlet form derived in Lemma 3, applied to the function $\langle w, \nabla_\theta \log p_\theta \rangle$, and ③ follows since $\Gamma_{MLE} = \left[ \mathbb{E}_p \nabla_\theta \log p_\theta \nabla_\theta \log p_\theta^\top \right]^{-1}$ (i.e. the inverse Fisher matrix (Van der Vaart, 2000)). Since this holds for every vector $w$, we have $\nabla_\theta^2 D_{GSM} \succeq \frac{1}{C_P} \Gamma_{MLE}^{-1}$. By monotonicity of the matrix inverse operator (Toda, 2011), the claim of the lemma follows. $\square$

# 4 BENEFITS OF ANNEALING: CONTINUOUSLY TEMPERED LANGEVIN DYNAMICS

In this section, we show another technique used to speed up Markov chains: lifting the Markov chain by introducing additional variables (e.g., momentum in underdamped Langevin, temperature in tempering techniques) can be used to design better score losses to deal with multimodality in the data distribution. Precisely, we introduce a diffusion we term *Continuously Tempered Langevin Dynamics*, which is a close relative of simulated tempering (Marinari & Parisi, 1992), where the number of "temperatures" is infinite, and we temper by convolving with Gaussian noise. We show that the generalized score matching loss corresponding to this Markov process mixes in time $\text{poly}(D, d)$ for a mixture of $K$ Gaussians (with identical covariance) in $d$ dimensions, and means in a ball of radius $D$. More precisely, in this section, we will consider the following family of distributions:

**Assumption 1.** *Let $p_0$ be a $d$-dimensional Gaussian distribution with mean 0 and covariance $\Sigma$. We will assume the data distribution $p$ is a $K$-Gaussian mixture, namely $p = \sum_{i=1}^K w_i p_i$, where*

$p_i(x) = p_0(x - \mu_i)$, *i.e. a shift of the distribution $p_0$ so its mean is $\mu_i$. We will assume the means $\mu_i$ lie within a ball with diameter $D$. We will denote the min and max eigenvalues of covariance with $\lambda_{\min}(\Sigma) = \lambda_{\min}$ and $\lambda_{\max}(\Sigma) = \lambda_{\max}$. We will denote the min and max mixture proportion with $\min_i w_i = w_{\min}$ and $\max_i w_i = w_{\max}$. Let $\Sigma_\beta = \Sigma + \beta\lambda_{\min}I_d$ be the shorthand notation of the covariance of individual Gaussian at temperature $\beta$.*

Mixtures of Gaussians are one of the most classical distributions in statistics—and they have very rich modeling properties. They are universal approximators in the sense that any distribution can be approximated (to any desired accuracy), if we consider a mixture with sufficiently many components (Alspach & Sorenson, 1972). A mixture of $K$ Gaussians is also the prototypical example of a distribution with $K$ modes — the shape of which is determined by the covariance of the components. Note at we are just saying that the data distribution $p$ can be described as a mixture of Gaussians, we are not saying anything about the parametric family we are fitting when optimizing the score matching loss—we need not necessarily fit the natural unknown parameters (the means, covariances and weights).

The primary reason this family of distributions is convenient for technical analysis is a closure property under convolutions: a convolution of a Gaussian mixture with a Gaussian produces another Gaussian mixture. Namely, the following holds from the distributivity property of the convolution operator, which is due to the linearity of an integral:

**Proposition 1** (Convolution with Gaussian). *Under Assumption 1, the distribution $p * \mathcal{N}(x; 0, \sigma^2 I)$ satisfies $p * \mathcal{N}(x; 0, \sigma^2 I) = \sum_i w_i \left( p_0(x - \mu_i) * \mathcal{N}(x; 0, \sigma^2 I) \right)$ and $(p_0(x - \mu_i) * \mathcal{N}(x; 0, \sigma^2 I))$ is a multivariate Gaussian with mean $\mu_i$ and covariance $\Sigma + \sigma^2 I$.*

The Markov process we will be analyzing (and the corresponding score matching loss) is a continuous-time analog of the Simulated Tempering Langevin Monte Carlo chain introduced in Ge et al. (2018):

**Definition 7** (Continuously Tempered Langevin Dynamics (CTLD)). *We will consider an SDE over a temperature-augmented state space, that is a random variable $(X_t, \beta_t), X_t \in \mathbb{R}^d, \beta_t \in \mathbb{R}^+$, defined as*

$$\begin{cases} dX_t = \nabla_x \log p^\beta(X_t)dt + \sqrt{2}dB_t \\ d\beta_t = \nabla_\beta \log r(\beta_t)dt + \nabla_\beta \log p^\beta(X_t)dt + \nu_t L(dt) + \sqrt{2}dB_t \end{cases}$$

*where $r : [0, \beta_{\max}] \to \mathbb{R}$ denotes the distribution over $\beta$, $r(\beta) \propto \exp\left(-\frac{7D^2}{\lambda_{\min}(1+\beta)}\right)$ and $\beta_{\max} = \frac{14D^2}{\lambda_{\min}} - 1$. Let $p^\beta := p * \mathcal{N}(0, \beta\lambda_{\min}I_d)$ denotes the distribution $p$ convolved with a Gaussian of covariance $\beta\lambda_{\min}I_d$. Furthermore, $L(dt)$ is a measure supported on the boundary of the interval $[0, \beta_{\max}]$ and $\nu_t$ is the unit normal at the endpoints of the interval, such that the stationary distribution of this SDE is $p(x, \beta) = r(\beta)p^\beta(x)$ (Saisho, 1987).*

**Remark 4.** *The above process can be readily seen as a "continuous-time" analogue of the usual simulated tempering chain (Lee et al., 2018; Ge et al., 2018), which either evolves $x$ according to a Markov chain with probability $p^\beta$, or changes $\beta$ (which has a discrete number of possible values), and applies an appropriate Metropolis-Hastings filter. The stationary distribution is $p(x, \beta) = r(\beta)p^\beta(x)$, since the updates amount to performing (reflected) Langevin dynamics corresponding to this stationary distribution.*

**Remark 5.** *The existence of the boundary measure is a standard result of reflecting diffusion processes via solutions to the Skorokhod problem (Saisho, 1987). If we ignore the boundary reflection term, the updates for CTLD are simply Langevin dynamics applied to the distribution $p(x, \beta)$. $r(\beta)$ specifies the distribution over the different levels of noise and is set up roughly so the Gaussians in the mixture have variance $\beta\Sigma$ with probability $\exp(-\Theta(\beta))$.*

Since CTLD amounts to performing (reflected) Langevin dynamics on the appropriate joint distribution $p(x, \beta)$, the corresponding generator $\mathcal{L}$ for CTLD is also readily written down:

**Proposition 2** (Dirichlet form for CTLD). *The Dirichlet form corresponding to CTLD has the form*

$$\mathcal{E}(f(x, \beta)) = \mathbb{E}_{p(x,\beta)}\|\nabla f(x, \beta)\|^2 = \mathbb{E}_{r(\beta)}\mathcal{E}_\beta(f(\cdot, \beta)) \tag{9}$$

*where $\mathcal{E}_\beta$ is the Dirichlet form corresponding to the Langevin diffusion (Lemma 3) with stationary distribution $p(x|\beta)$.*

Next, we derive the explicit score loss due to CTLD:

**Proposition 3.** *The generalized score matching loss with $\mathcal{O} = \nabla_{x,\beta}$ satisfies*

$$\left[\nabla_\theta^2 D_{GSM}(p, p_{\theta^*})\right]^{-1} \preceq C_P \Gamma_{MLE}$$

*Moreover,*

$$D_{GSM}(p, p_\theta) = \mathbb{E}_{\beta \sim r(\beta)} \mathbb{E}_{x \sim p^\beta} (\|\nabla_x \log p(x, \beta) - \nabla_x \log p_\theta(x, \beta)\|^2 + \|\nabla_\beta \log p(x, \beta) - \nabla_\beta \log p_\theta(x, \beta)\|^2)$$

$$= \mathbb{E}_{\beta \sim r(\beta)} \mathbb{E}_{x \sim p^\beta} \|\nabla_x \log p(x|\beta) - \nabla_x \log p_\theta(x|\beta)\|^2$$

$$+ \lambda_{\min} \mathbb{E}_{\beta \sim r(\beta)} \mathbb{E}_{x \sim p^\beta} \left( \left( \operatorname{Tr} \nabla_x^2 \log p(x|\beta) - \operatorname{Tr} \nabla_x^2 \log p_\theta(x|\beta) \right) + \left( \|\nabla_x \log p(x|\beta)\|_2^2 - \|\nabla_x \log p_\theta(x|\beta)\|_2^2 \right) \right)^2$$

*Proof.* The first claim follows by Lemma 4 as a special case of Langevin on the lifted distribution. The second claim follows by writing $\nabla_\beta \log p(x|\beta)$ and $\nabla_\beta \log p_\theta(x|\beta)$ through the Fokker-Planck equation for $p(x|\beta)$ (see Lemma 15). $\qquad\square$

This loss was derived from first principles from the Markov Chain-based framework in Section 3, however, it is readily seen that this loss is a "second-order" version of the annealed losses in Song & Ermon (2019); Song et al. (2020) — the weights being given by the distribution $r(\beta)$. Additionally, this loss has terms matching "second order" behavior of the distributions, namely $\operatorname{Tr} \nabla_x^2 \log p(x|\beta)$ and $\|\nabla_x \log p(x|\beta)\|_2^2$ with a weighting of $\lambda_{\min}$. Note this loss would be straightforward to train by the change of variables formula (Proposition 6, Appendix D)—and we also note that somewhat related "higher-order" analogues of score matching have appeared in the literature (without analysis or guarantees), for example, Meng et al. (2021).

The main result on the mixing time of CTLD is the following:

**Theorem 3** (Poincaré constant of CTLD)**.** *Under Assumption 1, the Poincaré constant of CTLD $C_P$ enjoys the upper bound $C_P \lesssim D^{22} d^2 \lambda_{\max}^9 \lambda_{\min}^{-2}$.*

Note that the above result has *no dependence* on the number of components, or on the smallest component weight $w_{\min}$—only on the diameter $D$, the ambient dimension $d$, and $\lambda_{\min}$ and $\lambda_{\max}$. This result thus applies to very general distributions, intuitively having an arbitrary number of modes which lie in a ball of radius $D$, and a bound on their "peakiness" and "spread". Our bounds are very likely not tight: our main thrust is to prove a polynomial dependence on the relevant parameters, which standard Langevin dynamics cannot achieve.

To get a bound on the asymptotic sample complexity of generalized score matching, according to the framework from Theorem 2, we also need to bound the smoothness terms (Lemma 14 in the general framework). These terms of course depend on the choice of parametrization for the family of distributions we are fitting. To get a quantitative sense for how these terms might scale, we will consider the natural parametrization for a mixture:

**Assumption 2.** *Consider the case of learning unknown means, such that the parameters to be learned are a vector $\theta = (\mu_1, \mu_2, \ldots, \mu_K) \in \mathbb{R}^{dK}$.*

Note that in this parametrization, we assume that the weights $\{w_i\}_{i=1}^K$ and shared covariance matrix $\Sigma$ are known, though the results can be straightforwardly generalized to the natural parametrization in which we are additionally fitting a vector $\{w_i\}_{i=1}^K$ and matrix $\Sigma$, at the expense of some calculational complexity. With this parametrization, the smoothness term can be bounded as follows:

**Theorem 4** (Smoothness under the natural parameterization)**.** *Under Assumptions 1 and 2, the following upper bound obtains:*

$$\|cov\left(\nabla_\theta \nabla_{x,\beta} \log p_\theta^\top \nabla_{x,\beta} \log p_\theta\right)\|_{OP} + \|cov\left(\nabla_\theta \Delta_{x,\beta} \log p_\theta\right)\|_{OP} \lesssim \operatorname{poly}\left(D, d, \lambda_{\min}^{-1}\right)$$

Note the above result *also has no dependence* on the number of components, or on the smallest component weight $w_{\min}$. Finally, we show that the generalized score matching loss is asymptotically normal. The proof of this is in Appendix F, and proceeds by verifying standard technical conditions for asymptotic behavior of M-estimators (Lemma 5), along with the Poincaré inequality bound in Theorem 3 and the framework in Theorem 2. As in Theorem 2, $n$ will denote the number of samples, and $\hat{\mathbb{E}}$ will denote an empirical average, that is the expectation over the $n$ training samples. We show:

**Theorem 5** (Main, Polynomial Sample Complexity Bound of CTLD)**.** *Let the data distribution $p$ satisfy Assumption 1. Then, the generalized score matching loss defined in Proposition 6 with parametrization as in Assumption 2 satisfies:*

1. *The set of optima $\Theta^* := \{\theta^* = (\mu_1, \mu_2, \ldots, \mu_K) | D_{GSM}(p, p_{\theta^*}) = \min_\theta D_{GSM}(p, p_\theta)\}$ satisfies:*

   $$\theta^* = (\mu_1, \mu_2, \ldots, \mu_K) \in \Theta^* \text{ if and only if } \exists \pi : [K] \to [K] \text{ satisfying } \forall i \in [K], \mu_{\pi(i)} = \mu_i^*, w_{\pi(i)} = w_i\}$$

2. *Let $\theta^* \in \Theta^*$ and let $C$ be any compact set containing $\theta^*$. Denote $C_0 = \{\theta \in C : p_\theta(x) = p(x) \text{ almost everywhere }\}$. Finally, let $D$ be any closed subset of $C$ not intersecting $C_0$. Then, we have $\lim_{n\to\infty} \Pr\left[\inf_{\theta \in D} \widehat{D_{GSM}}(\theta) < \widehat{D_{GSM}}(\theta^*)\right] \to 0$.*

3. *For every $\theta^* \in \Theta^*$ and every sufficiently small neighborhood $\hat{S}$ of $\theta^*$, there exists a sufficiently large $n$, such that there is a unique minimizer $\hat{\theta}_n$ of $\hat{\mathbb{E}} l_\theta(x)$ in $S$. Furthermore, $\hat{\theta}_n$ satisfies: $\sqrt{n}(\hat{\theta}_n - \theta^*) \xrightarrow{d} \mathcal{N}(0, \Gamma_{SM})$ for a matrix $\Gamma_{SM}$ satisfying $\|\Gamma_{SM}\|_{OP} \leq \text{poly}\left(D, d, \lambda_{\max}, \lambda_{\min}^{-1}\right) \|\Gamma_{MLE}\|_{OP}^2$.*

We provide some brief comments on each parts of this theorem. The first condition is the standard identifiability condition (Yakowitz & Spragins, 1968) for mixtures of Gaussians: the means are identifiable up to "renaming" the components. This is of course, inevitable if some of the weights are equal; if all the weights are distinct, $\Theta^*$ would in fact only consist of one point, s.t. $\forall i \in [K], \mu_i = \mu_i^*$. The second condition says that asymptotically, the empirical minimizers of $D_{GSM}$ are the points in $\Theta^*$. It can be viewed as (and follows from) a uniform law of large numbers. Finally, the third point characterizes the sample complexity of minimizers in the neirhborhood of each of the points in $\Theta^*$, and is a consequence of the CTLD Poincaré inequality estimate (Theorem 3) and the smoothness estimate (Theorem 4). Note that in fact the RHS of point 3 has *no dependence* on the number of components. This makes the result extremely general: the loss compared to MLE is very mild even for distributions with a large number of modes. [3]

**Bounding the Poincaré constant:** We will first sketch the proof of Theorem 3. By slight abuse of notation, we will define the distribution of the "individual components" of the mixture at a particular temperature, namely for $i \in [K]$, define $p(x, \beta, i) = r(\beta) w_i \mathcal{N}(x; \mu_i, \Sigma + \beta \lambda_{\min} I_d)$. Correspondingly, we will denote the conditional distribution for the $i$-th component by $p(x, \beta|i) \propto r(\beta) \mathcal{N}(x; \mu_i, \Sigma + \beta \lambda_{\min} I_d)$. The proof proceeds by applying the decomposition Theorem 6 to CTLD. Towards that, we denote by $\mathcal{E}_i$ the Dirichlet form corresponding to Langevin with stationary distribution $p(x, \beta|i)$. By Proposition 2, it's easy to see that the generator for CTLD satisfies $\mathcal{E} = \sum_i w_i \mathcal{E}_i$. This verifies condition (1) in Theorem 6. To verify condition (2), we will show Langevin for each of the distributions $p(x, \beta|i)$ mixes fast (i.e. the Poincaré constant is bounded). The details of this are provided in Section E.1. To verify condition (3), we will show the projected chain "between" the components (as defined in Theorem 6) mixes fast. The details of this are provided in Section E.2.

**Smoothness under the natural parametrization:** To obtain the polynomial upper bound in Theorem 4, we note the two terms $\|\text{cov}\left(\nabla_\theta \nabla_{x,\beta} \log p_\theta^\top \nabla_{x,\beta} \log p_\theta\right)\|_{OP}$ and $\|\text{cov}\left(\nabla_\theta \Delta_{x,\beta} \log p_\theta\right)\|_{OP}$ can be completely characterized by bounds on the higher-order derivatives with respect to $x$ and $\mu_i$ of the log-pdf since derivatives with respect to $\beta$ can be related to derivatives with respect to $x$ via the Fokker-Planck equation (Lemma 15). The polynomial bound requires three ingredients: In Lemma 10, we relate the derivatives of the mixture to derivatives of components by recognizing the higher-order score functions (Janzamin et al., 2014) of the form $\frac{Dp}{p}$ is closely related to the convex perspective map. In Lemma 7, we derive a new result in mixed derivatives of Gaussian components based on Hermite polynomials. In Corollary 1, we handle log derivatives with higher-order versions of the Faá di Bruno formula (Constantine & Savits, 1996), which is a combinatorial formula characterizing higher-order analogues of the chain rule. See Appendix G for details.

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

# Part I

# Appendix

## Table of Contents

## A  PRELIMINARIES

### A.1  MARKOV CHAIN DECOMPOSITION THEOREMS

Our mixing time bounds for the Continuously Tempered Langevin Dynamics will heavily use decomposition theorems to bound the Poincaré constant. These results "decompose" the Markov chain by partitioning the state space into sets, such that: (1) the mixing time of the Markov chain inside the sets is good; (2) the "projected" chain, which transitions between sets with probability equal to the probability flow between sets, also mixes fast.

In particular, we recall the following two results:

**Theorem 6** (Decomposition of Markov Chains, Theorem 6.1 in Ge et al. (2018)). *Let $M = (\Omega, \mathcal{L})$ be a continuous-time Markov chain with stationary distribution $p$ and Dirichlet form $\mathcal{E}(g, g) = -\langle g, \mathcal{L}g \rangle_p$. Suppose the following hold.*

*1. The Dirichlet form for $\mathcal{L}$ decomposes as $\langle f, \mathcal{L}g \rangle_p = \sum_{j=1}^m w_j \langle f, \mathcal{L}_j g \rangle_{p_j}$, where*

$$p = \sum_{j=1}^m w_j p_j$$

*and $\mathcal{L}_j$ is the generator for some Markov chain $M_j$ on $\Omega$ with stationary distribution $p_j$.*

*2. (Mixing for each $M_j$) The Dirichlet form $\mathcal{E}_j(f, g) = -\langle f, \mathcal{L}g \rangle_{p_j}$ satisfies the Poincaré inequality*

$$\mathrm{Var}_{p_j}(g) \leq C\mathcal{E}_j(g, g).$$

*3. (Mixing for projected chain) Define the $\chi^2$-projected chain $\bar{M}$ as the Markov chain on $[m]$ generated by $\bar{\mathcal{L}}$, where $\bar{\mathcal{L}}$ acts on $g \in L^2([m])$ by*

$$\bar{\mathcal{L}}\bar{g}(j) = \sum_{1 \leq k \leq m, k \neq j} [\bar{g}(k) - \bar{g}(j)]\bar{P}(j, k), \ \text{where} \ \bar{P}(j, k) = \frac{w_k}{\max\{\chi^2(p_j, p_k), \chi^2(p_k, p_j), 1\}}.$$

*Let $\bar{p}$ be the stationary distribution of $\bar{M}$. Suppose $\bar{M}$ satisfies the Poincaré inequality $\mathrm{Var}_{\bar{p}}(\bar{g}) \leq \bar{C}\bar{\mathcal{E}}(g, g)$.*

*Then $M$ satisfies the Poincaré inequality*

$$\mathrm{Var}_p(g) \leq C \left(1 + \frac{\bar{C}}{2}\right) \mathcal{E}(g, g).$$

The Poincaré constant bounds we will prove will also use a "continuous" version of the decomposition Theorem 6, which also appeared in Ge et al. (2018):

**Theorem 7** (Continuous decomposition theorem, Theorem D.3 in Ge et al. (2018)). *Consider a probability measure $\pi$ with $C^1$ density on $\Omega = \Omega^{(1)} \times \Omega^{(2)}$, where $\Omega^{(1)} \subseteq \mathbb{R}^{d_1}$ and $\Omega^{(2)} \subseteq \mathbb{R}^{d_2}$ are closed sets. For $X = (X_1, X_2) \sim P$ with probability density function $p$ (i.e., $P(dx) = p(x)\,dx$ and $P(dx_2|x_1) = p(x_2|x_1)\,dx_2$), suppose that*

- *The marginal distribution of $X_1$ satisfies a Poincaré inequality with constant $C_1$.*

- *For any $x_1 \in \Omega^{(1)}$, the conditional distribution $X_2|X_1 = x_1$ satisfies a Poincaré inequality with constant $C_2$.*

*Then $\pi$ satisfies a Poincaré inequality with constant*

$$\tilde{C} = \max\left\{C_2 \left(1 + 2C_1 \left\|\int_{\Omega^{(2)}} \frac{\|\nabla_{x_1}p(x_2|x_1)\|^2}{p(x_2|x_1)}dx_2\right\|_{L^\infty(\Omega^{(1)})}\right), 2C_1\right\}$$

## A.2 ASYMPTOTIC EFFICIENCY

We will need a classical result about asymptotic convergence of M-estimators, under some mild identifiability and differentiability conditions. For this section, $n$ will denote the number of samples, and $\hat{\mathbb{E}}$ will denote an empirical average, that is the expectation over the $n$ training samples. The following result holds:

**Lemma 5** (Van der Vaart (2000), Theorem 5.23). *Consider a loss $L : \Theta \mapsto \mathbb{R}$, such that $L(\theta) = \mathbb{E}_p[\ell_\theta(x)]$ for $l_\theta : \mathcal{X} \mapsto \mathbb{R}$. Let $\Theta^*$ be the set of global minima of $L$, that is*

$$\Theta^* = \{\theta^* : L(\theta^*) = \min_{\theta \in \Theta} L(\theta)\}$$

*Suppose the following conditions are met:*

- *(Gradient bounds on $l_\theta$) The map $\theta \mapsto l_\theta(x)$ is measurable and differentiable at every $\theta^* \in \Theta^*$ for $p$-almost every $x$. Furthermore, there exists a function $B(x)$, s.t. $\mathbb{E}B(x)^2 < \infty$ and for every $\theta_1, \theta_2$ near $\theta^*$, we have:*

$$|l_{\theta_1}(x) - l_{\theta_2}(x)| < B(x)\|\theta_1 - \theta_2\|_2$$

- *(Twice-differentiability of L) $L(\theta)$ is twice-differentiable at every $\theta^* \in \Theta^*$ with Hessian $\nabla^2_\theta L(\theta^*)$, and furthermore $\nabla^2_\theta L(\theta^*) \succ 0$.*

- *(Uniform law of large numbers) The loss $L$ satisfies a uniform law of large numbers, that is*

$$\sup_{\theta \in \Theta} \left|\hat{\mathbb{E}}l_\theta(x) - L(\theta)\right| \xrightarrow{p} 0$$

*Then, for every $\theta^* \in \Theta^*$, and every sufficiently small neighborhood $S$ of $\theta^*$, there exists a sufficiently large $n$, such that there is a unique minimizer $\hat{\theta}_n$ of $\hat{\mathbb{E}}l_\theta(x)$ in $S$. Furthermore, $\hat{\theta}_n$ satisfies:*

$$\sqrt{n}(\hat{\theta}_n - \theta^*) \xrightarrow{d} \mathcal{N}\left(0, (\nabla^2_\theta L(\theta^*))^{-1} Cov(\nabla_\theta \ell(x; \theta^*))(\nabla^2_\theta L(\theta^*))^{-1}\right)$$

## A.3 HERMITE POLYNOMIALS

To obtain polynomial bounds on the moments of derivatives of Gaussians, we will use the known results on multivariate Hermite polynomials.

**Definition 8** (Hermite polynomial, (Holmquist, 1996)). *The multivariate Hermite polynomial of order $k$ corresponding to a Gaussian with mean 0 and covariance $\Sigma$ is given by the Rodrigues formula:*

$$H_k(x; \Sigma) = (-1)^k \frac{(\Sigma \nabla_x)^{\otimes k} \phi(x; \Sigma)}{\phi(x; \Sigma)}$$

*where $\phi(x; \Sigma)$ is the pdf of a $d$-variate Gaussian with mean 0 and covariance $\Sigma$, and $\otimes$ denotes the Kronecker product.*

Note that $\nabla^{\otimes k}_x$ can be viewed as a formal Kronecker product, so that $\nabla^{\otimes k}_x f(x)$, where $f : \mathbb{R}^d \to \mathbb{R}$ is a $C^k$-smooth function gives a $d^k$-dimensional vector consisting of all partial derivatives of $f$ of order up to $k$.

**Proposition 4** (Integral representation of Hermite polynomial, (Holmquist, 1996)). *The Hermite polynomial $H_k$ defined in Definition 8 satisfies the integral formula:*

$$H_k(x; \Sigma) = \int (x + iu)^{\otimes k} \phi(u; \Sigma) du$$

*where $\phi(x; \Sigma)$ is the pdf of a $d$-variate Gaussian with mean 0 and covariance $\Sigma$.*

Note, the Hermite polynomials are either even functions or odd functions, depending on whether $k$ is even or odd:

$$H_k(-x; \Sigma) = (-1)^k H_k(x; \Sigma) \tag{10}$$

This property can be observed from the Rodrigues formula, the fact that $\phi(\cdot; \Sigma)$ is symmetric around 0, and the fact that $\nabla_{-x} = -\nabla_x$.

We establish the following relationship between Hermite polynomial and (potentially mixed) derivatives in $x$ and $\mu$, which we will use to bound several smoothness terms appearing in Section G.

**Lemma 6.** *If $\phi(x; \Sigma)$ is the pdf of a d-variate Gaussian with mean $0$ and covariance $\Sigma$, we have:*

$$\frac{\nabla_\mu^{k_1} \nabla_x^{k_2} \phi(x - \mu; \Sigma)}{\phi(x - \mu; \Sigma)} = (-1)^{k_2} \mathbb{E}_{u \sim \mathcal{N}(0, \Sigma)}[\Sigma^{-1}(x - \mu + iu)]^{\otimes(k_1 + k_2)}$$

*where the left-hand-side is understood to be shaped as a vector of dimension $\mathbb{R}^{d^{k_1 + k_2}}$.*

*Proof.* Using the fact that $\nabla_{x - \mu} = \nabla_x$ in Definition 8, we get:

$$H_k(x - \mu; \Sigma) = (-1)^k \frac{(\Sigma \nabla_x)^{\otimes k} \phi(x - \mu; \Sigma)}{\phi(x - \mu; \Sigma)}$$

Since the Kronecker product satisfies the property $(A \otimes B)(C \otimes D) = (AC) \otimes (BD)$, we have $(\Sigma \nabla_x)^{\otimes k} = \Sigma^{\otimes k} \nabla_x^{\otimes k}$. Thus, we have:

$$\frac{\nabla_x^k \phi(x - \mu; \Sigma)}{\phi(x - \mu; \Sigma)} = (-1)^k (\Sigma^{-1})^{\otimes k} H_k(x - \mu; \Sigma) \tag{11}$$

Since $\phi(\mu - x; \Sigma)$ is symmetric in $\mu$ and $x$, taking derivatives with respect to $\mu$ we get:

$$H_k(\mu - x; \Sigma) = (-1)^k \frac{(\Sigma \nabla_\mu)^k \phi(\mu - x; \Sigma)}{\phi(\mu - x; \Sigma)}$$

Rearranging again and using (10), we get:

$$\frac{\nabla_\mu^k \phi(x - \mu; \Sigma)}{\phi(x - \mu; \Sigma)} = (\Sigma^{-1})^{\otimes k} H_k(x - \mu; \Sigma) \tag{12}$$

Combining (11) and (12), we get:

$$\begin{aligned}
\frac{\nabla_\mu^{k_1} \nabla_x^{k_2} \phi(x - \mu; \Sigma)}{\phi(x - \mu; \Sigma)} &= (-1)^{k_2} \frac{\nabla_\mu^{k_1}[(\Sigma^{-1})^{\otimes k_2} H_{k_2}(x - \mu; \Sigma) \phi(x - \mu; \Sigma)]}{\phi(x - \mu; \Sigma)} \\
&= (-1)^{k_2} \frac{\nabla_\mu^{k_1}[\nabla_\mu^{k_2} \phi(x - \mu; \Sigma)]}{\phi(x - \mu; \Sigma)} \\
&= (-1)^{k_2} \frac{\nabla_\mu^{k_1 + k_2} \phi(x - \mu; \Sigma)}{\phi(x - \mu; \Sigma)} \\
&= (-1)^{k_2} (\Sigma^{-1})^{\otimes(k_1 + k_2)} H_{k_1 + k_2}(x - \mu; \Sigma)
\end{aligned}$$

Applying the integral formula from Proposition 4, we have:

$$\frac{\nabla_\mu^{k_1} \nabla_x^{k_2} \phi(x - \mu; \Sigma)}{\phi(x - \mu; \Sigma)} = (-1)^{k_2} \int [\Sigma^{-1}(x - \mu + iu)]^{\otimes(k_1 + k_2)} \phi(u; \Sigma) \, du$$

as we needed. $\square$

Now we are ready to obtain an explicit polynomial bound for the mixed derivatives for a multivariate Gaussian with mean $\mu$ and covariance $\Sigma$. We have the following bounds:

**Lemma 7.** *If $\phi(x; \Sigma)$ is the pdf of a d-variate Gaussian with mean $0$ and covariance $\Sigma$, we have:*

$$\left\| \frac{\nabla_\mu^{k_1} \nabla_x^{k_2} \phi(x - \mu; \Sigma)}{\phi(x - \mu; \Sigma)} \right\|_2 \lesssim \|\Sigma^{-1}(x - \mu)\|_2^{k_1 + k_2} + d^{(k_1 + k_2)/2} \lambda_{\min}^{-(k_1 + k_2)/2}$$

*where the left-hand-side is understood to be shaped as a vector of dimension $\mathbb{R}^{d^{k_1 + k_2}}$.*

*Proof.* We start with Lemma 6 and use the convexity of the norm

$$\left\| \frac{\nabla_\mu^{k_1} \nabla_x^{k_2} \phi(x - \mu; \Sigma)}{\phi(x - \mu; \Sigma)} \right\|_2 \leq \mathbb{E}_{u \sim \mathcal{N}(0, \Sigma)} \|[\Sigma^{-1}(x - \mu + iu)]^{\otimes(k_1 + k_2)}\|_2$$

Bounding the right-hand side, we have:

$$\mathbb{E}_{u\sim\mathcal{N}(0,\Sigma)}\|[\Sigma^{-1}(x-\mu+iu)]^{\otimes(k_1+k_2)}\|_2 \lesssim \|\Sigma^{-1}(x-\mu)\|_2^{k_1+k_2} + \mathbb{E}_{u\sim\mathcal{N}(0,\Sigma)}\|\Sigma^{-1}u\|_2^{k_1+k_2}$$
$$= \|\Sigma^{-1}(x-\mu)\|_2^{k_1+k_2} + \mathbb{E}_{z\sim\mathcal{N}(0,I_d)}\|\Sigma^{-\frac{1}{2}}z\|_2^{k_1+k_2}$$
$$\leq \|\Sigma^{-1}(x-\mu)\|_2^{k_1+k_2} + \|\Sigma^{-\frac{1}{2}}\|_{OP}^{k_1+k_2}\mathbb{E}_{z\sim\mathcal{N}(0,I_d)}\|z\|_2^{k_1+k_2}$$

Applying Lemma 30 yields the desired result. □

Similarly, we can bound mixed derivatives involving a Laplacian in $x$:

**Lemma 8.** *If* $\phi(x;\Sigma)$ *is the pdf of a* $d$*-variate Gaussian with mean* 0 *and covariance* $\Sigma$*, we have:*

$$\left\|\frac{\nabla_\mu^{k_1}\Delta_x^{k_2}\phi(x-\mu;\Sigma)}{\phi(x-\mu;\Sigma)}\right\| \lesssim \sqrt{d^{k_2}}\|\Sigma^{-1}(x-\mu)\|_2^{k_1+2k_2} + d^{(k_1+3k_2)/2}\lambda_{\min}^{-(k_1+2k_2)/2}$$

*Proof.* By the definition of a Laplacian, and the AM-GM inequality, we have, for any function $f:\mathbb{R}^d\to\mathbb{R}$

$$(\Delta^k f(x))^2 = \left(\sum_{i_1,i_2,\ldots,i_k=1}^d \partial_{i_1}^2\partial_{i_2}^2\cdots\partial_{i_k}^2 f(x)\right)^2$$
$$\leq d^k \sum_{i_1,i_2,\ldots,i_k=1}^d \left(\partial_{i_1}^2\partial_{i_2}^2\cdots\partial_{i_k}^2 f(x)\right)^2$$
$$\leq d^k \|\nabla_x^{2k} f(x)\|_2^2$$

Thus, we have

$$\left\|\frac{\nabla_\mu^{k_1}\Delta_x^{k_2}\phi(x-\mu;\Sigma)}{\phi(x-\mu;\Sigma)}\right\|_2 \leq \sqrt{d^{k_2}}\left\|\frac{\nabla_\mu^{k_1}\nabla_x^{2k_2}\phi(x-\mu;\Sigma)}{\phi(x-\mu;\Sigma)}\right\|_2$$

Applying Lemma 7, the result follows.

□

## A.4 LOGARITHMIC DERIVATIVES

Finally, we will need similar bounds for logarithic derivatives—that is, derivatives of $\log p(x)$, where $p$ is a multivariate Gaussian.

We recall the following result, which is a consequence of the multivariate extension of the Faá di Bruno formula:

**Proposition 5** (Constantine & Savits (1996), Corollary 2.10)**.** *Consider a function* $f:\mathbb{R}^d\to\mathbb{R}$*, s.t.* $f$ *is* $N$ *times differentiable in an open neighborhood of* $x$ *and* $f(x)\neq 0$*. Then, for any multi-index* $I\in\mathbb{N}^d$*, s.t.* $|I|\leq N$*, we have:*

$$\partial_{x_I}\log f(x) = \sum_{k,s=1}^{|I|}\sum_{p_s(I,k)}(-1)^{k-1}(k-1)!\prod_{j=1}^s\frac{\partial_{l_j}f(x)^{m_j}}{f(x)^{m_j}}\frac{\prod_{i=1}^d(I_i)!}{m_j!l_j!^{m_j}}$$

*where* $p_s(I,k) = \{\{l_i\}_{i=1}^s \in (\mathbb{N}^d)^s, \{m_i\}_{i=1}^s \in \mathbb{N}^s : l_1 \prec l_2 \prec \cdots \prec l_s, \sum_{i=1}^s m_i = k, \sum_{i=1}^s m_i l_i = I\}$*.*

*The* $\prec$ *ordering on multi-indices is defined as follows:* $(a_1,a_2,\ldots,a_d) := a \prec b := (b_1,b_2,\ldots,b_d)$ *if:*

1. $|a| < |b|$

2. $|a| = |b|$ *and* $a_1 < b_1$*.*

3. $|a| = |b|$ and $\exists k >= 1$, s.t. $\forall j \leq k, a_j = b_j$ and $a_{k+1} < b_{k+1}$.

As a straightforward corollary, we have the following:

**Corollary 1.** *For any multi-index $I \in \mathbb{N}^d$, s.t. $|I|$ is a constant, we have*

$$|\partial_{x_I} \log f(x)| \lesssim \max\left(1, \max_{J \leq I}\left|\frac{\partial_J f(x)}{f(x)}\right|^{|I|}\right)$$

*where $J \in \mathbb{N}^d$ is a multi-index, and $J \leq I$ iff $\forall i \in d, J_i \leq I_i$.*

### A.5 MOMENTS OF MIXTURES AND THE PERSPECTIVE MAP

The main strategy in bounding moments of quantities involving a mixture will be to leverage the relationship between the expectation of the score function and the so-called *perspective map*. In particular, this allows us to bound the moments of derivatives of the mixture score in terms of those of the individual component scores, which are easier to bound using the machinery of Hermite polynomials in the prior section.

Note in this section all derivatives are calculated at $\theta = \theta^*$ and therefore $p(x, \beta) = p_\theta(x, \beta)$.

**Lemma 9.** *(Convexity of perspective, [Boyd & Vandenberghe (2004)](#)) Let $f$ be a convex function. Then, its corresponding perspective map $g(u, v) := vf\left(\frac{u}{v}\right)$ with domain $\{(u, v) : \frac{u}{v} \in Dom(f), v > 0\}$ is convex.*

We will apply the following lemma many times, with appropriate choice of differentiation operator $D$ and power $k$.

**Lemma 10.** *Let $D : \mathcal{F}^1 \to \mathcal{F}^m$ be a linear operator that maps from the space of all scalar-valued functions to the space of m-variate functions of $x \in \mathbb{R}^d$ and let $\theta$ be such that $p = p_\theta$. For $k \in \mathbb{N}$, and any norm $\|\cdot\|$ of interest*

$$\mathbb{E}_{(x,\beta)\sim p(x,\beta)}\left\|\frac{(Dp_\theta)(x|\beta)}{p_\theta(x|\beta)}\right\|^k \leq \max_{\beta,i}\mathbb{E}_{x\sim p(x|\beta,i)}\left\|\frac{(Dp_\theta)(x|\beta,i)}{p_\theta(x|\beta,i)}\right\|^k$$

*Proof.* Let us denote $g(u, v) := v\left\|\frac{u}{v}\right\|^k$. Note that since any norm is convex by definition, so is $g$, by Lemma 9. Then, we proceed as follows:

$$\mathbb{E}_{(x,\beta)\sim p(x,\beta)}\left\|\frac{(Dp_\theta)(x|\beta)}{p_\theta(x|\beta)}\right\|^k = \mathbb{E}_{\beta\sim r(\beta)}\mathbb{E}_{x\sim p(x|\beta)}\left\|\frac{(Dp_\theta)(x|\beta)}{p_\theta(x|\beta)}\right\|^k$$

$$= \mathbb{E}_{\beta\sim r(\beta)}\int g((Dp_\theta)(x|\beta), p_\theta(x|\beta))dx$$

$$= \mathbb{E}_{\beta\sim r(\beta)}\int g\left(\sum_{i=1}^{K}w_i(Dp_\theta)(x|\beta,i), \sum_{i=1}^{K}w_ip_\theta(x|\beta,i)\right)dx \quad (13)$$

$$\leq \mathbb{E}_{\beta\sim r(\beta)}\int \sum_{i=1}^{K}w_ig((Dp_\theta)(x|\beta,i), p_\theta(x|\beta,i))dx \quad (14)$$

$$= \mathbb{E}_{\beta\sim r(\beta)}\sum_{i=1}^{K}w_i\mathbb{E}_{x\sim p(x|\beta,i)}\left\|\frac{(Dp_\theta)(x|\beta,i)}{p_\theta(x|\beta,i)}\right\|^k$$

$$\leq \max_{\beta,i}\mathbb{E}_{x\sim p(x|\beta,i)}\left\|\frac{(Dp_\theta)(x|\beta,i)}{p_\theta(x|\beta,i)}\right\|^k$$

where (13) follows by linearity of $D$, and (14) by convexity of the function $g$. $\qquad\square$

## B    GENERATORS AND SCORE LOSSES FOR DIFFUSIONS

In this section, we derive several expressions about generators, Dirichlet forms, and associated generalized score matching losses for diffusions of the kind (7).

First, we derive the Dirichlet form of Itô diffusions of the form (7). Namely, we show:

**Lemma 11** (Dirichlet form of continuous Markov Process). *Suppose $p$ vanishes at infinity. For an Itô diffusion of the form* (7)*, its Dirichlet form is:*

$$\mathcal{E}(g) = \mathbb{E}_p \| \sqrt{D(x)} \nabla g(x) \|_2^2$$

*Proof.*  By Itô's Lemma, the generator $\mathcal{L}$ of the Itô diffusion (7) is:

$$(\mathcal{L}g)(x) = \langle -[D(x) + Q(x)]\nabla f(x) + \Gamma(x), \nabla g(x) \rangle + \mathrm{Tr}(D(x)\nabla^2 g(x))$$

The Dirichlet form is given by

$$\mathcal{E}(g) = -\mathbb{E}_p \langle \mathcal{L}g, g \rangle \quad = -\int p(x) \left[ \underbrace{\langle -[D(x) + Q(x)]\nabla f(x) + \Gamma(x), \nabla g(x) \rangle}_{\text{I}} + \underbrace{\mathrm{Tr}(D(x)\nabla^2 g(x))}_{\text{II}} \right] g(x)dx$$

Expanding and using the definition of $\Gamma$, term I can be written as:

$$\text{I} = \int p(x) \langle D(x)\nabla f(x), \nabla g(x) \rangle g(x)dx \tag{15}$$

$$+ \int p(x) \langle Q(x)\nabla_x f(x), \nabla g(x) \rangle g(x)dx \tag{16}$$

$$- \int p(x) \sum_{i,j} \partial_j D_{ij}(x) \partial_i g(x) g(x)dx \tag{17}$$

$$- \int p(x) \sum_{i,j} \partial_j Q_{ij}(x) \partial_i g(x) g(x)dx \tag{18}$$

We will simplify term II via a sequence of integration by parts:

$$\text{II} = -\int p(x) \mathrm{Tr}(D(x)\nabla^2 g(x)) g(x)dx$$

$$= -\int p(x) \left( \sum_{i,j} D_{ij}(x) \partial_{ij} g(x) \right) g(x)dx$$

$$= -\sum_{i,j} \int p(x) D_{ij}(x) g(x) \partial_{ij} g(x)dx$$

$$= -\sum_{i,j} \left( p(x)D_{ij}(x)g(x)\partial_i g(x) \Big|_{x=-\infty}^{\infty} - \int \partial_j[p(x)D_{ij}(x)g(x)]\partial_i g(x)dx \right)$$

$$= \sum_{i,j} \int \partial_j[p(x)D_{ij}(x)g(x)]\partial_i g(x)dx$$

$$= \sum_{i,j} \int \partial_j p(x) D_{ij}(x) g(x) \partial_i g(x)dx \tag{19}$$

$$+ \sum_{i,j} \int p(x)\partial_j D_{ij}(x) g(x) \partial_i g(x)dx \tag{20}$$

$$+ \sum_{i,j} \int p(x)D_{ij}(x)\partial_j g(x) \partial_i g(x)dx \tag{21}$$

The term (19) cancels out with term (15), so that we get:

$$\sum_{i,j} \int \partial_j p(x) D_{ij}(x) g(x) \partial_i g(x) dx$$

$$= \sum_{i,j} \int p(x) \partial_j \log p(x) D_{ij}(x) g(x) \partial_i g(x) dx$$

$$= -\int p(x) \langle D(x) \nabla_x f(x), \nabla_x g(x) \rangle g(x) dx$$

The term (20) cancels out with the term (17).

For term (16),

$$\int p(x) \langle Q(x) \nabla_x f(x), \nabla_x g(x) \rangle g(x) dx$$

$$= -\int \langle Q(x) \nabla_x p(x), \nabla_x g(x) \rangle g(x) dx$$

$$= \int \langle \nabla_x p(x), Q(x) \nabla_x g(x) \rangle g(x) dx$$

$$= \int \sum_{i,j} \partial_j p(x) Q_{ji}(x) \partial_i g(x) g(x) dx$$

$$= -\int \sum_{i,j} \partial_j p(x) Q_{ij}(x) \partial_i g(x) g(x) dx$$

Combining term (16) and term (18),

$$\int p(x) \langle Q(x) \nabla_x f(x), \nabla_x g(x) \rangle g(x) dx - \int p(x) \sum_{i,j} \partial_j Q_{ij}(x) \partial_i g(x) g(x) dx$$

$$= -\int \sum_{i,j} [\partial_j p(x) Q_{ij}(x) + p(x) \partial_j Q_{ij}(x)] \partial_i g(x) g(x) dx$$

$$= -\sum_{i,j} \int \partial_j [p(x) Q_{ij}(x)] \partial_i g(x) g(x) dx$$

$$= -\sum_{i,j} \left( p(x) Q_{ij}(x) \partial_i g(x) g(x) \Big|_{x=-\infty}^{\infty} - \int p(x) Q_{ij}(x) \partial_j [\partial_i g(x) g(x)] dx \right)$$

$$= \sum_{i,j} \int p(x) Q_{ij}(x) [\partial_{ij} g(x) g(x) + \partial_i g(x) \partial_j g(x)] dx$$

$$= \frac{1}{2} \sum_{i,j} \int p(x) \{ Q_{ij}(x) [\partial_{ij} g(x) g(x) + \partial_i g(x) \partial_j g(x)] + Q_{ji}(x) [\partial_{ji} g(x) g(x) + \partial_j g(x) \partial_i g(x)] \} dx$$

$$= \frac{1}{2} \sum_{i,j} \int p(x) \{ Q_{ij}(x) [\partial_{ij} g(x) g(x) + \partial_i g(x) \partial_j g(x)] - Q_{ij}(x) [\partial_{ji} g(x) g(x) + \partial_j g(x) \partial_i g(x)] \} dx$$

$$= 0$$

In the end, we are only left with term (21):

$$\mathcal{E}(g) = \sum_{i,j} \int p(x) D_{ij}(x) \partial_j g(x) \partial_i g(x) dx$$

$$= \int p(x) \langle \nabla_x g(x), D(x) \nabla_x g(x) \rangle dx$$

$$= \mathbb{E}_p \| \sqrt{D(x)} \nabla_x g(x) \|_2^2$$

□

We also calculate the integration by parts version of the generalized score matching loss for (4).

**Lemma 12** (Integration by parts for the GSM in (4)). *Suppose $p$ vanishes at infinity. The generalized score matching objective in (4) satisfies the equality*

$$D_{GSM}(p, q) = \frac{1}{2}\left[\mathbb{E}_p\|B(x)\nabla \log q\|^2 + 2\mathbb{E}_p div\left(B(x)^2\nabla \log q\right)\right] + K_p$$

*Proof.* Expanding the squares in (4), we have:

$$D_{GSM}(p, q) = \frac{1}{2}\left[\mathbb{E}_p\|B(x)\nabla \log p\|^2 + \mathbb{E}_p\|B(x)\nabla \log q\|^2 - 2\mathbb{E}_p\langle B(x)\nabla \log p, B(x)\nabla \log q\rangle\right]$$

The cross-term can be rewritten using integration by parts as:

$$\mathbb{E}_p\langle B(x)\nabla \log p, B(x)\nabla \log q\rangle = \int_x \langle \nabla p, B(x)^2\nabla \log q\rangle$$
$$= -\int_x p(x)\text{div}\left(B(x)^2\nabla \log q\right)$$
$$= -\mathbb{E}_p\text{div}\left(B(x)^2\nabla \log q\right)$$

□

## C  A FRAMEWORK FOR ANALYZING GENERALIZED SCORE MATCHING

**Lemma 13** (Hessian of GSM loss). *The Hessian of $D_{GSM}$ defined in Theorem 2 satisfies*

$$\nabla_\theta^2 D_{GSM}(p, p_{\theta^*}) = \mathbb{E}_p\left[\nabla_\theta\nabla_x \log p_{\theta^*}(x)^\top D(x)\nabla_\theta\nabla_x \log p_{\theta^*}(x)\right]$$

*Proof.* By a straightforward calculation, we have:

$$\nabla_\theta D_{GSM}(p, p_\theta) = \mathbb{E}_p\nabla_\theta\left(\frac{\sqrt{D(x)}\nabla_x p_\theta(x)}{p_\theta(x)}\right)\left(\frac{\sqrt{D(x)}\nabla_x p_\theta(x)}{p_\theta(x)} - \frac{\sqrt{D(x)}\nabla_x p(x)}{p(x)}\right)$$

$$\nabla_\theta^2 D_{GSM}(p, p_\theta) = \mathbb{E}_p\nabla_\theta\left(\frac{\sqrt{D(x)}\nabla_x p_\theta(x)}{p_\theta(x)}\right)^\top\nabla_\theta\left(\frac{\sqrt{D(x)}\nabla_x p_\theta(x)}{p_\theta(x)}\right)$$
$$- \left(\frac{\sqrt{D(x)}\nabla_x p_\theta(x)}{p_\theta(x)} - \frac{\sqrt{D(x)}\nabla_x p(x)}{p(x)}\right)^\top\nabla_\theta^2\left(\frac{\sqrt{D(x)}\nabla_x p_\theta(x)}{p_\theta(x)}\right)$$

Since $\frac{\sqrt{D(x)}\nabla_x p_{\theta^*}(x)}{p_{\theta^*}(x)} = \frac{\sqrt{D(x)}\nabla_x p(x)}{p(x)}$, the second term vanishes at $\theta = \theta^*$.

$$\nabla_\theta^2 D_{GSM}(p, p_{\theta^*}) = \mathbb{E}_p\left[\nabla_\theta\left(\frac{\sqrt{D(x)}\nabla_x p_{\theta^*}(x)}{p_{\theta^*}(x)}\right)^\top\nabla_\theta\left(\frac{\sqrt{D(x)}\nabla_x p_{\theta^*}(x)}{p_{\theta^*}(x)}\right)\right]$$

□

**Lemma 14** (Bound on smoothness). *For $l_\theta(x)$ defined in Theorem 2,*

$$cov(\nabla_\theta l_\theta(x)) \precsim cov\left(\nabla_\theta\nabla_x \log p_\theta(x)D(x)\nabla_x \log p_\theta(x)\right)$$
$$+ cov\left(\nabla_\theta\nabla_x \log p_\theta(x)^\top div(D(x))\right)$$
$$+ cov\left(\nabla_\theta \text{Tr}[D(x)\Delta \log p_\theta(x)]\right)$$

*Proof.* We have

$$\nabla_\theta l_\theta(x) = \frac{1}{2}\nabla_\theta \left[\|\sqrt{D(x)}\nabla_x \log p_\theta(x)\|^2 + 2\mathrm{div}\left(D(x)\nabla_x \log p_\theta(x)\right)\right]$$

$$= \nabla_\theta \nabla_x \log p_\theta(x)D(x)\nabla_x \log p_\theta(x) + \nabla_\theta \nabla_x \log p_\theta(x)^\top \mathrm{div}(D(x)) + \nabla_\theta \mathrm{Tr}[D(x)\Delta \log p_\theta(x)]$$

By Lemma 2 in Koehler et al. (2022), we also have

$$\mathrm{cov}(\nabla_\theta l_\theta(x)) \precsim \mathrm{cov}\left(\nabla_\theta \nabla_x \log p_\theta(x)D(x)\nabla_x \log p_\theta(x)\right)$$
$$+ \mathrm{cov}\left(\nabla_\theta \nabla_x \log p_\theta(x)^\top \mathrm{div}(D(x))\right)$$
$$+ \mathrm{cov}\left(\nabla_\theta \mathrm{Tr}[D(x)\Delta \log p_\theta(x)]\right)$$

which completes the proof. $\qquad\square$

## D OVERVIEW OF CONTINUOUSLY TEMPERED LANGEVIN DYNAMICS

In this section, we provide several calculations around the score matching losses associated with Continuously Tempered Langevin Dynamics.

**Lemma 15** ($\beta$ derivatives via Fokker Planck)**.** *For any distribution $p^\beta$ such that $p^\beta = p * \mathcal{N}(0, \lambda_{\min}\beta I)$ for some $p$, we have the following PDE for its log-density:*

$$\nabla_\beta \log p^\beta(x) = \lambda_{\min}\left(\mathrm{Tr}\left(\nabla_x^2 \log p^\beta(x)\right) + \|\nabla_x \log p^\beta(x)\|_2^2\right)$$

*As a consequence, both $p(x|\beta, i)$ and $p(x|\beta)$ follow the above PDE.*

*Proof.* Consider the SDE $dX_t = \sqrt{2\lambda_{\min}}dB_t$. Let $q_t$ be the law of $X_t$. Then, $q_t = q_0 * N(0, \lambda_{\min}tI)$. On the other hand, by the Fokker-Planck equation, $\frac{d}{dt}q_t(x) = \lambda_{\min}\Delta_x q_t(x)$. From this, it follows that

$$\nabla_\beta p^\beta(x) = \lambda_{\min}\Delta_x p^\beta(x)$$
$$= \lambda_{\min}\mathrm{Tr}(\nabla_x^2 p^\beta(x))$$

Hence, by the chain rule,

$$\nabla_\beta \log p^\beta(x) = \frac{\lambda_{\min}\mathrm{Tr}(\nabla_x^2 p^\beta(x))}{p^\beta(x)} \tag{22}$$

Furthermore, by a straightforward calculation, we have

$$\nabla_x^2 \log p^\beta(x) = \frac{\nabla_x^2 p^\beta(x)}{p^\beta(x)} - \left(\nabla_x \log p^\beta(x)\right)\left(\nabla_x \log p^\beta(x)\right)^\top$$

Plugging this in (22), we have

$$\frac{\lambda_{\min}\mathrm{Tr}(\nabla_x^2 p^\beta(x))}{p^\beta(x)} = \lambda_{\min}\left(\mathrm{Tr}\left(\nabla_x^2 \log p^\beta(x)\right) + \mathrm{Tr}\left(\left(\nabla_x \log p^\beta(x)\right)\left(\nabla_x \log p^\beta(x)\right)^\top\right)\right)$$

$$= \lambda_{\min}\left(\mathrm{Tr}\left(\nabla_x^2 \log p^\beta(x)\right) + \mathrm{Tr}\left(\left(\nabla_x \log p^\beta(x)\right)^\top\left(\nabla_x \log p^\beta(x)\right)\right)\right)$$

$$= \lambda_{\min}\left(\mathrm{Tr}\left(\nabla_x^2 \log p^\beta(x)\right) + \|\nabla_x \log p^\beta(x)\|_2^2\right)$$

as we needed. $\qquad\square$

**Proposition 6** (Integration-by-part Generalized Score Matching Loss for CTLD)**.** *The loss $D_{GSM}$ in the integration by parts form (Lemma 1) as $D_{GSM}(p, p_\theta) = \mathbb{E}_p l_\theta(x, \beta) + K_p$, where*

$$l_\theta(x, \beta) := l_\theta^1(x, \beta) + l_\theta^2(x, \beta), \text{ and } l_\theta^1(x, \beta) := \frac{1}{2}\|\nabla_x \log p_\theta(x|\beta)\|_2^2 + \Delta_x \log p_\theta(x|\beta), \text{ and}$$

$$l_\theta^2(x, \beta) := \frac{1}{2}(\nabla_\beta \log p_\theta(x|\beta))^2 + \nabla_\beta \log r(\beta)\nabla_\beta \log p_\theta(x|\beta) + \Delta_\beta \log p_\theta(x|\beta)$$

*Moreover, all the terms in the definition of $l_\theta^1(x, \beta)$ and $l_\theta^2(x, \beta)$ can be written as a sum of powers of partial derivatives of $\nabla_x \log p_\theta(x|\beta)$.*

*Proof of Lemma 6.*

$$D_{GSM}\left(p, p_\theta\right)$$

$$= \frac{1}{2}\mathbb{E}_p[\|\nabla_{(x,\beta)}\log p_\theta(x,\beta)\|_2^2 + 2\Delta_{(x,\beta)}\log p_\theta(x,\beta)]$$

$$= \frac{1}{2}\mathbb{E}_p[\|\nabla_x \log p_\theta(x,\beta)\|_2^2 + 2\Delta_x \log p_\theta(x,\beta) + \|\nabla_\beta \log p_\theta(x,\beta)\|_2^2 + 2\Delta_\beta \log p_\theta(x,\beta)]$$

$$= \frac{1}{2}\mathbb{E}_p[\|\nabla_x \log p_\theta(x|\beta) + \nabla_x \log r(\beta)\|_2^2 + 2\Delta_x \log p_\theta(x|\beta) + 2\Delta_x \log r(\beta)$$

$$+ \|\nabla_\beta \log p_\theta(x|\beta) + \nabla_\beta \log r(\beta)\|_2^2 + 2\Delta_\beta \log p_\theta(x|\beta) + 2\Delta_\beta \log r(\beta)]$$

$$= \mathbb{E}_p[\frac{1}{2}\|\nabla_x \log p_\theta(x|\beta)\|_2^2 + \Delta_x \log p_\theta(x|\beta)$$

$$+ \frac{1}{2}\|\nabla_\beta \log p_\theta(x|\beta)\|_2^2 + \nabla_\beta \log r(\beta)\nabla_\beta \log p_\theta(x|\beta) + \Delta_\beta \log p_\theta(x|\beta)] + C$$

By Lemma 15, $\nabla_\beta \log p_\theta(x|\beta)$ is a function of partial derivatives of the score $\nabla_x \log p_\theta(x|\beta)$. Similarly, $\nabla_\beta^2 \log p_\theta(x|\beta)$ can be shown to be a function of partial derivatives of the score $\nabla_x \log p_\theta(x|\beta)$ as well:

$$\Delta_\beta \log p_\theta(x|\beta) = \nabla_\beta \lambda_{\min}(\text{Tr}(\nabla_x^2 \log p_\theta(x|\beta)) + \|\nabla_x \log p_\theta(x|\beta)\|_2^2)$$

$$= \lambda_{\min}(\text{Tr}(\nabla_x^2 \nabla_\beta \log p_\theta(x|\beta)) + 2\nabla_x\nabla_\beta \log p_\theta(x|\beta)^\top \nabla_x \log p_\theta(x|\beta))$$

$\square$

## E  POLYNOMIAL MIXING TIME BOUND: PROOF OF THEOREM 3

The proof of Theorem 3 will follow by applying Theorem 6. Towards that, we need to verify the three conditions of the theorem:

1. (Decomposition of Dirichlet form) The Dirichlet energy of CTLD for $p(x,\beta)$, by the tower rule of expectation, decomposes into a linear combination of the Dirichlet forms of Langevin with stationary distribution $p(x,\beta|i)$. Precisely, we have

$$\mathbb{E}_{(x,\beta)\sim p(x,\beta)}\|\nabla f(x,\beta)\|^2 = \sum_i w_i \mathbb{E}_{(x,\beta)\sim p(x,\beta|i)}\|\nabla f(x,\beta)\|^2$$

2. (Polynomial mixing for individual modes) By Lemma 16, for all $i \in [K]$ the distribution $p(x,\beta|i)$ has Poincaré constant $C_{x,\beta|i}$ with respect to the Langevin generator that satisfies:

$$C_{x,\beta|i} \lesssim D^{20}d^2\lambda_{\max}^9\lambda_{\min}^{-1}$$

3. (Polynomial mixing for projected chain) To bound the Poincaré constant of the projected chain, by Lemma 19 we have

$$\bar{C} \lesssim D^2\lambda_{\min}^{-1}$$

Putting the above together, by Theorem 6.1 in Ge et al. (2018) we have:

$$C_P \leq C_{x,\beta|i}\left(1 + \frac{\bar{C}}{2}\right)$$

$$\leq C_{x,\beta|i}\bar{C}$$

$$\lesssim D^{22}d^2\lambda_{\max}^9\lambda_{\min}^{-2}$$

### E.1  FAST MIXING WITHIN A COMPONENT

The first claim we will show is that we have fast mixing "inside" each of the components of the mixture. Formally, we show:

**Lemma 16.** *For $i \in [K]$, let $C_{x,\beta|i}$ be the Poincaré constant of $p(x, \beta|i)$. Then, we have $C_{x,\beta|i} \lesssim D^{20} d^2 \lambda_{\max}^9 \lambda_{\min}^{-1}$.*

The proof of this lemma proceeds via another (continuous) decomposition theorem. Intuitively, what we show is that for every $\beta$, $p(x|\beta, i)$ has a good Poincaré constant; moreover, the marginal distribution of $\beta$, which is $r(\beta)$, is log-concave and supported over a convex set (an interval), so has a good Poincaré constant. Putting these two facts together via a continuous decomposition theorem (Theorem D.3 in Ge et al. (2018), repeated as Theorem 7), we get the claim of the lemma.

*Proof.* The proof will follow by an application of a continuous decomposition result (Theorem D.3 in Ge et al. (2018), repeated as Theorem 7) , which requires three bounds:

1. A bound on the Poincaré constants of the distributions $p(\beta|i)$: since $\beta$ is independent of $i$, we have $p(\beta|i) = r(\beta)$. Since $r(\beta)$ is a log-concave distribution over a convex set (an interval), we can bound its Poincaré constant by standard results (Bebendorf, 2003). The details are in Lemma 17, $C_\beta \leq \frac{14D^2}{\pi \lambda_{\min}}$.

2. A bound on the Poincaré constant $C_{x|\beta,i}$ of the conditional distribution $p(x|\beta, i)$: We claim $C_{x|\beta,i} \leq \lambda_{\max} + \beta \lambda_{\min}$. This follows from standard results on Poincaré inequalities for strongly log-concave distributions. Namely, by the Bakry-Emery criterion, an $\alpha$-strongly log-concave distribution has Poincaré constant $\frac{1}{\alpha}$ (Bakry & Émery, 2006). Since $p(x|\beta, i)$ is a Gaussian whose covariance matrix has smallest eigenvalue lower bounded by $\lambda_{\max} + \beta \lambda_{\min}$, it is $(\lambda_{\max} + \beta \lambda_{\min})^{-1}$-strongly log-concave. Since $\beta \in [0, \beta_{\max}]$, we have $C_{x|\beta,i} \leq \lambda_{\max} + \beta_{\max} \lambda_{\min} \leq \lambda_{\max} + 14D^2$.

3. A bound on the "rate of change" of the density $p(x|\beta, i)$, i.e. $\left\| \int \frac{\|\nabla_\beta p(x|\beta,i)\|_2^2}{p(x|\beta,i)} dx \right\|_{L^\infty}$: This is done via an explicit calculation, the details of which are in Lemma 18.

By Theorem D.3 in Ge et al. (2018), the Poincaré constant $C_{x,\beta|i}$ of $p(x, \beta|i)$ enjoys the upper bound:

$$C_{x,\beta|i} \leq \max \left\{ C_{x|\beta_{\max},i} \left( 1 + C_\beta \left\| \int \frac{\|\nabla_\beta p(x|\beta,i)\|_2^2}{p(x|\beta,i)} dx \right\|_{L^\infty(\beta)} \right), 2C_\beta \right\}$$

$$\lesssim \max \left\{ \left( \lambda_{\max} + 14D^2 \right) \left( 1 + \frac{14D^2}{\pi \lambda_{\min}} d^2 \max\{\lambda_{\max}^8, D^{16}\} \right), \frac{28D^2}{\pi \lambda_{\min}} \right\}$$

$$\lesssim \frac{D^{20} d^2 \lambda_{\max}^9}{\lambda_{\min}}$$

which completes the proof. $\qquad\square$

**Lemma 17** (Bound on the Poincaré constant of $r(\beta)$)**.** *Let $C_\beta$ be the Poincaré constant of the distribution $r(\beta)$ with respect to reflected Langevin diffusion. Then,*

$$C_\beta \leq \frac{14D^2}{\pi \lambda_{\min}}$$

*Proof.* We first show that $r(\beta)$ is a log-concave distribution. By a direct calculation, the second derivative in $\beta$ satisfies:

$$\nabla_\beta^2 \log r(\beta) = -\frac{14D^2}{\lambda_{\min}(1 + \beta)^3} \leq 0$$

Since the interval is a convex set, with diameter $\beta_{\max}$, by Bebendorf (2003) we have

$$C_\beta \leq \frac{\beta_{\max}}{\pi} = \frac{14D^2}{\pi \lambda_{\min}} - \frac{1}{\pi}$$

from which the Lemma immediately follows. $\qquad\square$

**Lemma 18** (Bound on "rate of change" of the density $p(x|\beta, i)$)**.**

$$\left\| \int \frac{\|\nabla_\beta p(x|\beta, i)\|_2^2}{p(x|\beta, i)} dx \right\|_{L^\infty(\beta)} \lesssim d^2 \max\{\lambda_{\max}^8, D^{16}\}$$

*Proof.*

$$\left\| \int \frac{\|\nabla_\beta p(x|\beta, i)\|_2^2}{p(x|\beta, i)} dx \right\|_{L^\infty(\beta)} = \left\| \int \left(\nabla_\beta \log p(x|\beta, i)\right)^2 p(x|\beta, i) dx \right\|_{L^\infty(\beta)}$$

$$= \sup_\beta \mathbb{E}_{x \sim p(x|\beta, i)} \left(\nabla_\beta \log p(x|\beta, i)\right)^2$$

We can apply Lemma 15 to derive explicit expressions for the right-hand side:

$$\left\| \int \frac{\|\nabla_\beta p(x|\beta, i)\|_2^2}{p(x|\beta, i)} dx \right\|_{L^\infty(\beta)} = \sup_\beta \mathbb{E}_{x \sim p(x|\beta, i)} \lambda_{\min}^2 \left[\text{Tr}(\Sigma_\beta^{-1}) + \|\Sigma_\beta(x - \mu_i)\|_2^2\right]^2$$

$$\overset{①}{\leq} 2\lambda_{\min}^2 \sup_\beta \left[\text{Tr}(\Sigma_\beta^{-1})^2 + \mathbb{E}_{x \sim p(x|\beta, i)}\|\Sigma_\beta(x - \mu_i)\|_2^4\right]$$

$$\leq 2\lambda_{\min}^2 \sup_\beta \left[d^2((1 + \beta)\lambda_{\min})^{-2} + \mathbb{E}_{z \sim \mathcal{N}(0,I)}\|\Sigma_\beta^{\frac{3}{2}} z \Sigma_\beta^{\frac{1}{2}}\|_2^4\right]$$

$$\leq 2\lambda_{\min}^2 \sup_\beta \left[d^2((1 + \beta)\lambda_{\min})^{-2} + \|\Sigma_\beta^{\frac{3}{2}}\|_{OP}^4 \|\Sigma_\beta^{\frac{1}{2}}\|_{OP}^4 \mathbb{E}_{z \sim \mathcal{N}(0,I)}\|z\|_2^4\right]$$

$$\overset{②}{\leq} 4 \sup_\beta \left[d^2(1 + \beta)^{-2} + \lambda_{\min}^2 \|\Sigma_\beta\|_{OP}^8 d^2\right]$$

$$= 4 \sup_\beta \left[d^2(1 + \beta)^{-2} + \lambda_{\min}^2 (\lambda_{\max} + \beta\lambda_{\min})^8 d^2\right]$$

$$= 4 \left(d^2 + \lambda_{\min}^2 (\lambda_{\max} + \beta_{\max}\lambda_{\min})^8 d^2\right)$$

$$\overset{③}{\leq} 4d^2 + 4d^2 \lambda_{\min}^2 (\lambda_{\max} + 14D^2)^8$$

$$\leq 16d^2 \max\{\lambda_{\max}^8, 14^8 D^{16}\}$$

In ①, we use $(a + b)^2 \leq 2(a^2 + b^2)$ for $a, b \geq 0$; in ② we apply the moment bound for the Chi-Squared distribution of degree-of-freedom $d$ in Lemma 30; and in ③ we plug in the bound on $\beta_{\max}$. □

### E.2 MIXING BETWEEN COMPONENTS

Next, we show the "projected" chain between the components mixes fast:

**Lemma 19** (Poincaré constant of projected chain)**.** *Define the projected chain $\bar{M}$ over $[K]$ with transition probability*

$$T(i, j) = \frac{w_j}{\max\{\chi_{\max}^2(p(x, \beta|i), p(x, \beta|j)), 1\}}$$

*where $\chi_{\max}^2(p, q) = \max\{\chi^2(p, q), \chi^2(q, p)\}$. If $\sum_{j \neq i} T(i, j) < 1$, the remaining mass is assigned to the self-loop $T(i, i)$. The stationary distribution $\bar{p}$ of this chain satisfies $\bar{p}(i) = w_i$. Furthermore, the projected chain has Poincaré constant*

$$\bar{C} \lesssim D^2 \lambda_{\min}^{-1}.$$

The intuition for this claim is that the transition probability graph is complete, i.e. $T(i, j) \neq 0$ for every pair $i, j \in [K]$. Moreover, the transition probabilities are lower bounded, since the $\chi^2$ distances

between any pair of "annealed" distributions $p(x, \beta|i)$ and $p(x, \beta|j)$ can be upper bounded. The reason for this is that at large $\beta$, the Gaussians with mean $\mu_i$ and $\mu_j$ are smoothed enough so that they have substantial overlap; moreover, the distribution $r(\beta)$ is set up so that enough mass is placed on the large $\beta$. The precise lemma bounding the $\chi^2$ divergence between the components is stated as Lemma 20.

*Proof.* The stationary distribution follows from the detailed balance condition $w_i T(i, j) = w_j T(j, i)$.

We upper bound the Poincaré constant using the method of canonical paths (Diaconis & Stroock, 1991). For all $i, j \in [K]$, we set $\gamma_{ij} = \{(i, j)\}$ to be the canonical path. Define the weighted length of the path

$$\begin{aligned}
\|\gamma_{ij}\|_T &= \sum_{(k,l) \in \gamma_{ij}, k, l \in [K]} T(k, l)^{-1} \\
&= T(i, j)^{-1} \\
&= \frac{\max\{\chi^2_{\max}(p(x, \beta|i), p(x, \beta|j)), 1\}}{w_j} \\
&\leq \frac{14 D^2}{\lambda_{\min} w_j}
\end{aligned}$$

where the inequality comes from Lemma 20 which provides an upper bound for the chi-squared divergence. Since $D$ is an upper bound and $\lambda_{\min}$ is a lower bound, we may assume without loss of generality that $\chi^2_{\max} \geq 1$.

Finally, we can upper bound the Poincaré constant using Proposition 1 in Diaconis & Stroock (1991)

$$\begin{aligned}
\bar{C} &\leq \max_{k,l \in [K]} \sum_{\gamma_{ij} \ni (k,l)} \|\gamma_{ij}\|_T w_i w_j \\
&= \max_{k,l \in [K]} \|\gamma_{kl}\|_T w_k w_l \\
&\leq \frac{14 D^2 w_{\max}}{\lambda_{\min}} \\
&\leq \frac{14 D^2}{\lambda_{\min}}
\end{aligned}$$

$\square$

Next, we will prove a bound on the chi-square distance between the joint distributions $p(x, \beta|i)$ and $p(x, \beta|j)$. Intuitively, this bound is proven by showing bounds on the chi-square distances between $p(x|\beta, i)$ and $p(x|\beta, j)$ (Lemma 21) — which can be explicitly calculated since they are Gaussian, along with tracking how much weight $r(\beta)$ places on each of the $\beta$. Moreover, the Gaussians are flatter for larger $\beta$, so they overlap more — making the chi-square distance smaller.

**Lemma 20** ($\chi^2$-divergence between joint "annealed" Gaussians)**.**

$$\chi^2(p(x, \beta|i), p(x, \beta|j)) \leq \frac{14 D^2}{\lambda_{\min}}$$

*Proof.* Expanding the definition of $\chi^2$-divergence, we have:

$$
\begin{aligned}
\chi^2(p(x,\beta|i), p(x,\beta|j)) &= \int \left( \frac{p(x,\beta|i)}{p(x,\beta|j)} - 1 \right)^2 p(x,\beta|i) dx d\beta \\
&= \int_0^{\beta_{\max}} \int_{-\infty}^{+\infty} \left( \frac{p(x|\beta,i)r(\beta)}{p(x|\beta,j)r(\beta)} - 1 \right)^2 p(x|\beta,i)r(\beta) dx d\beta \\
&= \int_0^{\beta_{\max}} \chi^2(p(x|\beta,i), p(x|\beta,j)) r(\beta) d\beta \\
&\leq \int_0^{\beta_{\max}} \exp\left( \frac{7D^2}{\lambda_{\min}(1+\beta)} \right) r(\beta) d\beta \qquad (23) \\
&= \int_0^{\beta_{\max}} \exp\left( \frac{7D^2}{\lambda_{\min}(1+\beta)} \right) \frac{1}{Z(D, \lambda_{\min})} \exp\left( -\frac{7D^2}{\lambda_{\min}(1+\beta)} \right) d\beta \\
&= \frac{\beta_{\max}}{Z(D, \lambda_{\min})}
\end{aligned}
$$

where in Line 23, we apply our Lemma 21 to bound the $\chi^2$-divergence between two Gaussians with identical covariance. By a change of variable $\tilde\beta := \frac{7D^2}{\lambda_{\min}(1+\beta)}$, $\beta = \frac{7D^2}{\lambda_{\min}\tilde\beta} - 1$, $d\beta = -\frac{7D^2}{\lambda_{\min}} \frac{1}{\tilde\beta^2} d\tilde\beta$, we can rewrite the integral as:

$$
\begin{aligned}
Z(D, \lambda_{\min}) &= \int_0^{\beta_{\max}} \exp\left( -\frac{7D^2}{\lambda_{\min}(1+\beta)} \right) d\beta \\
&= -\frac{7D^2}{\lambda_{\min}} \int_{\frac{7D^2}{\lambda_{\min}}}^{\frac{7D^2}{\lambda_{\min}(1+\beta_{\max})}} \exp\left( -\tilde\beta \right) \frac{1}{\tilde\beta^2} d\tilde\beta \\
&= \frac{7D^2}{\lambda_{\min}} \int_{\frac{7D^2}{\lambda_{\min}(1+\beta_{\max})}}^{\frac{7D^2}{\lambda_{\min}}} \exp\left( -\tilde\beta \right) \frac{1}{\tilde\beta^2} d\tilde\beta \\
&\geq \frac{7D^2}{\lambda_{\min}} \int_{\frac{7D^2}{\lambda_{\min}(1+\beta_{\max})}}^{\frac{7D^2}{\lambda_{\min}}} \exp\left( -2\tilde\beta \right) d\tilde\beta \\
&= \frac{7D^2}{2\lambda_{\min}} \left( \exp\left( -\frac{14D^2}{\lambda_{\min}(1+\beta_{\max})} \right) - \exp\left( -\frac{14D^2}{\lambda_{\min}} \right) \right)
\end{aligned}
$$

Since $D$ is an upper bound and $\lambda_{\min}$ is a lower bound, we can assume $\frac{D^2}{\lambda_{\min}} \geq 1$ without loss of generality. Plugging in $\beta_{\max} = \frac{14D^2}{\lambda_{\min}} - 1$, we get

$$
Z(D, \lambda_{\min}) \geq \frac{7}{2} \left( \exp(-1) - \exp(-14) \right) \geq 1
$$

Finally, we get the desired bound

$$
\chi^2(p(x,\beta|i), p(x,\beta|j)) \leq \beta_{\max} = \frac{14D^2}{\lambda_{\min}} - 1
$$

$\square$

The next lemma bounds the $\chi^2$-divergence between two Gaussians with the same covariance.

**Lemma 21** ($\chi^2$-divergence between Gaussians with same covariance).

$$
\chi^2(p(x|\beta,i), p(x|\beta,j)) \leq \exp\left( \frac{7D^2}{\lambda_{\min}(1+\beta)} \right)
$$

*Proof.* Plugging in the definition of $\chi^2$-distance for Gaussians, we have:

$$\chi^2(p(x|\beta, i), p(x|\beta, j))$$

$$\leq \frac{\det(\Sigma_\beta)^{\frac{1}{2}}}{\det(\Sigma_\beta)} \det\left(\Sigma_\beta^{-1}\right)^{-\frac{1}{2}}$$

$$\exp\left(\frac{1}{2}\left(\Sigma_\beta^{-1}(2\mu_j - \mu_i)\right)^\top (\Sigma_\beta^{-1})^{-1}\left(\Sigma_\beta^{-1}(2\mu_j - \mu_i)\right) + \frac{1}{2}\mu_i^\top \Sigma_\beta^{-1}\mu_i - \mu_j^\top \Sigma_\beta^{-1}\mu_j\right) \quad (24)$$

$$= \exp\left(\frac{1}{2}\left(\Sigma_\beta^{-1}(2\mu_j - \mu_i)\right)^\top (\Sigma_\beta^{-1})^{-1}\left(\Sigma_\beta^{-1}(2\mu_j - \mu_i)\right) + \frac{1}{2}\mu_i^\top \Sigma_\beta^{-1}\mu_i\right)$$

$$\exp\left(-\mu_j^\top \Sigma_\beta^{-1}\mu_j\right)$$

$$\leq \exp\left(\frac{1}{2}(2\mu_j - \mu_i)^\top \Sigma_\beta^{-1}(2\mu_j - \mu_i) + \frac{1}{2}\mu_i^\top \Sigma_\beta^{-1}\mu_i\right) \quad (25)$$

$$\leq \exp\left(\frac{\|2\mu_j - \mu_i\|_2^2 + \|2\mu_i\|_2^2}{2\lambda_{\min}(1 + \beta)}\right)$$

$$\leq \exp\left(\frac{(\|2\mu_j\|_2 + \|\mu_i\|_2)^2 + 4\|\mu_i\|_2^2}{2\lambda_{\min}(1 + \beta)}\right)$$

$$\leq \exp\left(\frac{2\|2\mu_j\|_2^2 + 2\|\mu_i\|_2^2 + 4\|\mu_i\|_2^2}{2\lambda_{\min}(1 + \beta)}\right)$$

$$\leq \exp\left(\frac{7D^2}{\lambda_{\min}(1 + \beta)}\right)$$

In Equation 24, we apply Lemma G.7 from Ge et al. (2018) for the chi-square divergence between two Gaussian distributions. In Equation 25, we use the fact that $\Sigma_\beta^{-1}$ is PSD.

□

# F  ASYMPTOTIC NORMALITY OF GENERALIZED SCORE MATCHING FOR CTLD

The main theorem of this section is proving asymptotic normality for the generalized score matching loss corresponding to CTLD. Precisely, we show:

**Theorem 8** (Asymptotic normality of generalized score matching for CTLD). *Let the data distribution $p$ satisfy Assumption 1. Then, the generalized score matching loss defined in Proposition 6 satisfies:*

1. *The set of optima*

$$\Theta^* := \{\theta^* = (\mu_1, \mu_2, \ldots, \mu_K) | D_{GSM}(p, p_{\theta^*}) = \min_\theta D_{GSM}(p, p_\theta)\}$$

   *satisfies*

$$\theta^* = (\mu_1, \mu_2, \ldots, \mu_K) \in \Theta^* \text{ if and only if } \exists \pi : [K] \to [K] \text{ satisfying } \forall i \in [K], \mu_{\pi(i)} = \mu_i^*, w_{\pi(i)} = w_i\}$$

2. *Let $\theta^* \in \Theta^*$ and let $C$ be any compact set containing $\theta^*$. Denote*

$$C_0 = \{\theta \in C : p_\theta(x) = p(x) \text{ almost everywhere }\}$$

   *Finally, let $D$ be any closed subset of $C$ not intersecting $C_0$. Then, we have:*

$$\lim_{n \to \infty} Pr\left[\inf_{\theta \in D} \widehat{D_{GSM}}(\theta) < \widehat{D_{GSM}}(\theta^*)\right] \to 0$$

3. *For every $\theta^* \in \Theta^*$ and every sufficiently small neighborhood $S$ of $\theta^*$, there exists a sufficiently large $n$, such that there is a unique minimizer $\hat{\theta}_n$ of $\hat{\mathbb{E}}l_\theta(x)$ in $S$. Furthermore, $\hat{\theta}_n$ satisfies:*

$$\sqrt{n}(\hat{\theta}_n - \theta^*) \xrightarrow{d} \mathcal{N}(0, \Gamma_{SM})$$

   *for some matrix $\Gamma_{SM}$.*

*Proof.* Part 1 is shown in Lemma 22: the claim roughly follows by classic results on the identifiability of the parameters of a mixture (up to permutations of the components) (Yakowitz & Spragins, 1968).

Part 2 is shown in Lemma 24: it follows from a uniform law of large numbers.

Finally, Part 3 follows from an application of Lemma 5—so we verify the conditions of the lemma are satisfied. The gradient bounds on $l_\theta$ are verified Lemma 23—and it largely follows by moment bounds on gradients of the score derived in Section G. Uniform law of large numbers is shown in Lemma 24, and the the existence of Hessian of $L = D_{GSM}$ is trivially verified. $\square$

For the sake of notational brevity, in this section, we will slightly abuse notation and denote $D_{GSM}(\theta) := D_{GSM}(p, p_\theta)$.

**Lemma 22** (Uniqueness of optima). *Suppose for $\theta := (\mu_1, \mu_2, \ldots, \mu_K)$ there is no permutation $\pi : [K] \to [K]$, such that $\mu_{\pi(i)} = \mu_i^*$ and $w_{\pi(i)} = w_i, \forall i \in [K]$. Then, $D_{GSM}(\theta) > D_{GSM}(\theta^*)$*

*Proof.* For notational convenience, let $D_{SM}$ denote the standard score matching loss, and let us denote $D_{SM}(\theta) := D_{SM}(p, p_\theta)$. For any distributions $p_\theta$, by Proposition 1 in Koehler et al. (2022), it holds that

$$D_{SM}(\theta) - D_{SM}(\theta^*) \geq \frac{1}{LSI(p_\theta)} \mathrm{KL}(p_{\theta^*}, p_\theta)$$

where $LSI(q)$ denotes the Log-Sobolev constant of the distribution $q$. If $\theta = (\mu_1, \mu_2, \ldots, \mu_K)$ is such that there is no permutation $\pi : [K] \to [K]$ satisfying $\mu_{\pi(i)} = \mu_i^*$ and $w_{\pi(i)} = w_i, \forall i \in [K]$, by Yakowitz & Spragins (1968) we have $\mathrm{KL}(p_{\theta^*}, p_\theta) > 0$. Furthermore, the distribution $p_\theta$, by virtue of being a mixture of Gaussians, has a finite log-Sobolev constant (Theorem 1 in Chen et al. (2021)). Therefore, $D_{SM}(\theta) > D_{SM}(\theta^*)$.

However, note that $D_{GSM}(p_\theta)$ is a (weighted) average of $D_{SM}$ losses, treating the data distribution as $p_{\theta^*}^\beta$, a convolution of $p_{\theta^*}$ with a Gaussian with covariance $\beta\lambda_{\min}I_d$; and the distribution being fitted as $p_\theta^\beta$. Thus, the above argument implies that if $\theta \neq \theta^*$, we have $D_{GSM}(\theta) > D_{GSM}(\theta^*)$, as we need. $\square$

**Lemma 23** (Gradient bounds of $l_\theta$). *Let $l_\theta(x, \beta)$ be as defined in Proposition 6. Then, there exists a constant $C(d, D, \frac{1}{\lambda_{\min}})$ (depending on $d, D, \frac{1}{\lambda_{\min}}$), such that*

$$\mathbb{E}\|\nabla_\theta l(x, \beta)\|^2 \leq C\left(d, D, \frac{1}{\lambda_{\min}}\right)$$

*Proof.* By Proposition 6,

$$l_\theta(x, \beta) = l_\theta^1(x, \beta) + l_\theta^2(x, \beta), \text{ and}$$

$$l_\theta^1(x, \beta) := \frac{1}{2}\|\nabla_x \log p_\theta(x|\beta)\|_2^2 + \Delta_x \log p_\theta(x|\beta)$$

$$l_\theta^2(x, \beta) := \frac{1}{2}(\nabla_\beta \log p_\theta(x|\beta))^2 + \nabla_\beta \log r(\beta)\nabla_\beta \log p_\theta(x|\beta) + \Delta_\beta \log p_\theta(x|\beta)$$

Using repeatedly the fact that $\|a + b\|^2 \leq 2\left(\|a\|^2 + \|b\|^2\right)$, we have:

$$\mathbb{E}\|l_\theta(x, \beta)\|_2^2 \lesssim \mathbb{E}\left\|l_\theta^2(x, \beta)\right\|_2^2 + \mathbb{E}\left\|l_\theta^2(x, \beta)\right\|_2^2$$

$$\mathbb{E}\left\|l_\theta^1(x, \beta)\right\|_2^2 \lesssim \mathbb{E}\|\nabla_x \log p_\theta(x, \beta)\|_2^4 + \mathbb{E}\left(\Delta_x \log p_\theta(x, \beta)\right)^2$$

$$\mathbb{E}\left\|l_\theta^2(x, \beta)\right\|_2^2 \lesssim \mathbb{E}\left(\nabla_\beta \log p_\theta(x|\beta)\right)^4 + \mathbb{E}\left(\nabla_\beta \log r(\beta)\nabla_\beta \log p_\theta(x|\beta)\right)^2 + \mathbb{E}\left(\Delta_\beta \log p_\theta(x|\beta)\right)^2$$

We proceed to bound the right hand sides above. We have:

$$\mathbb{E}\left\|l_\theta^1(x, \beta)\right\|_2^2 \lesssim \mathbb{E}\|\nabla_x \log p_\theta(x, \beta)\|_2^4 + \mathbb{E}\left(\Delta_x \log p_\theta(x, \beta)\right)^2$$

$$\lesssim \max_{\beta,i}\mathbb{E}_{x \sim p(x|\beta,i)}\|\nabla_x \log p_\theta(x|\beta,i)\|_2^4 + \max_{\beta,i}\mathbb{E}_{x \sim p(x|\beta,i)}\left(\Delta_x \log p_\theta(x|\beta,i)\right)^2 \tag{26}$$

$$\leq \mathrm{poly}\left(d, \frac{1}{\lambda_{\min}}\right) \tag{27}$$

Where (26) follows by Lemma 10, and (27) follows by combining Corollaries 2 and 1.

The same argument, along with Lemma 15, and the fact that $\max_\beta (\nabla_\beta \log r(\beta))^4 \lesssim D^8 \lambda_{\min}^{-4}$ by a direct calculation shows that

$$\mathbb{E} \left\| l_\theta^2(x, \beta) \right\|_2^2 \lesssim \mathbb{E} \left( \nabla_\beta \log p_\theta(x|\beta) \right)^4 + \mathbb{E} \left( \nabla_\beta \log r(\beta) \nabla_\beta \log p_\theta(x|\beta) \right)^2 + \mathbb{E} \left( \Delta_\beta \log p_\theta(x|\beta) \right)^2$$

$$\leq \text{poly}\left( d, D, \frac{1}{\lambda_{\min}} \right)$$

$\square$

**Lemma 24** (Uniform convergence). *The generalized score matching loss satisfies a uniform law of large numbers:*

$$\sup_{\theta \in \Theta} \left| \widehat{D_{GSM}}(\theta) - D_{GSM}(\theta) \right| \xrightarrow{p} 0$$

*Proof.* The proof will proceed by a fairly standard argument, using symmetrization and covering number bounds. Precisely, let $T = \{(x_i, \beta_i)\}_{i=1}^n$ be the training data. We will denote by $\hat{\mathbb{E}}_T$ the empirical expectation (i.e. the average over) a training set $T$.

We will first show that

$$\mathbb{E}_T \sup_{\theta \in \Theta} \left| \widehat{D_{GSM}}(\theta) - D_{GSM}(\theta) \right| \leq \frac{C\left( K, d, D, \frac{1}{\lambda_{\min}} \right)}{\sqrt{n}} \tag{28}$$

from which the claim will follow. First, we will apply the symmetrization trick, by introducing a "ghost training set" $T' = \{(x_i', \beta_i')\}_{i=1}^n$. Precisely, we have:

$$\mathbb{E}_T \sup_{\theta \in \Theta} \left| \widehat{D_{GSM}}(\theta) - D_{GSM}(\theta) \right| = \mathbb{E}_T \sup_{\theta \in \Theta} \left| \hat{\mathbb{E}}_T l_\theta(x, \beta) - D_{GSM}(\theta) \right|$$

$$= \mathbb{E}_T \sup_{\theta \in \Theta} \left| \hat{\mathbb{E}}_T l_\theta(x, \beta) - \mathbb{E}_{T'} \hat{\mathbb{E}}_{T'} l_\theta(x, \beta) \right| \tag{29}$$

$$\leq \mathbb{E}_{T, T'} \sup_{\theta \in \Theta} \left| \frac{1}{n} \sum_{i=1}^n \left( l_\theta(x_i, \beta_i) - l_\theta(x_i', \beta_i') \right) \right| \tag{30}$$

where (29) follows by noting the population expectation can be expressed as the expectation over a choice of a (fresh) training set $T'$, (30) follows by applying Jensen's inequality. Next, consider Rademacher variables $\{\varepsilon_i\}_{i=1}^n$. Since a Rademacher random variable is symmetric about 0, we have

$$\mathbb{E}_{T, T'} \sup_{\theta \in \Theta} \left| \frac{1}{n} \sum_{i=1}^n \left( l_\theta(x_i, \beta_i) - l_\theta(x_i', \beta_i') \right) \right| = \mathbb{E}_{T, T'} \sup_{\theta \in \Theta} \left| \frac{1}{n} \sum_{i=1}^n \varepsilon_i \left( l_\theta(x_i, \beta_i) - l_\theta(x_i', \beta_i') \right) \right|$$

$$\leq 2 \mathbb{E}_T \sup_{\theta \in \Theta} \left| \frac{1}{n} \sum_{i=1}^n \varepsilon_i l_\theta(x_i, \beta_i) \right|$$

For notational convenience, let us denote by

$$R := \sqrt{\frac{1}{n} \sum_{i=1}^n \| \nabla_\theta l_\theta(x_i, \beta_i) \|^2}$$

We will bound this supremum by a Dudley integral, along with covering number bounds. Considering $T$ as fixed, with respect to the randomness in $\{\varepsilon_i\}$, the process $\frac{1}{n} \sum_{i=1}^n \varepsilon_i l_\theta(x_i, \beta_i)$ is subgaussian with respect to the metric

$$d(\theta, \theta') := \frac{1}{\sqrt{n}} R \|\theta - \theta'\|_2$$

In other words, we have

$$\mathbb{E}_{\{\varepsilon_i\}} \exp\left( \lambda \frac{1}{n} \sum_{i=1}^n \varepsilon_i \left( l_\theta(x_i, \beta_i) - l_{\theta'}(x_i, \beta_i) \right) \right) \leq \exp\left( \lambda^2 d(\theta, \theta') \right) \tag{31}$$

The proof of this is as follows: since $\varepsilon_i$ is 1-subgaussian, and

$$|l_\theta(x_i, \beta_i) - l_{\theta'}(x_i, \beta_i)| \leq \|\nabla_\theta l_\theta(x_i, \beta_i)\| \|\theta - \theta'\|$$

we have that $\varepsilon_i \left(l_\theta(x_i, \beta_i) - l_{\theta'}(x_i, \beta_i)\right)$ is subgaussian with variance proxy $\|\nabla_\theta(x_i, \beta_i)\|^2 \|\theta - \theta'\|^2$. Thus, $\frac{1}{n} \sum_{i=1}^n \varepsilon_i l_\theta(x_i, \beta_i)$ is subgaussian with variance proxy $\frac{1}{n^2} \sum_{i=1}^n \|\nabla_\theta l_\theta(x_i, \beta_i)\|^2 \|\theta - \theta'\|_2^2$, which is equivalent to (31).

The Dudley entropy integral then gives

$$\sup_{\theta \in \Theta} \left| \frac{1}{n} \sum_{i=1}^n \varepsilon_i l_\theta(x_i, \beta_i) \right| \lesssim \int_0^\infty \sqrt{\log N(\epsilon, \Theta, d)} d\epsilon \tag{32}$$

where $N(\epsilon, \Theta, d)$ denotes the size of the smallest possible $\epsilon$-cover of the set of parameters $\Theta$ in the metric $d$.

Note that the $\epsilon$ in the integral bigger than the diameter of $\Theta$ in the metric $d$ does not contribute to the integral, so we may assume the integral has an upper limit

$$M = \frac{2}{\sqrt{n}} RD$$

Moreover, $\Theta$ is a product of $K$ $d$-dimensional balls of (Euclidean) radius $D$, so

$$\log N(\epsilon, \Theta, d) \leq \log\left(\left(1 + \frac{RD}{\sqrt{n}\epsilon}\right)^{Kd}\right)$$

$$\leq \frac{KdRD}{\sqrt{n}\epsilon}$$

Plugging this estimate back in (32), we get

$$\sup_{\theta \in \Theta} \left| \frac{1}{n} \sum_{i=1}^n \varepsilon_i l_\theta(x_i, \beta_i) \right| \lesssim \sqrt{KdRD/\sqrt{n}} \int_0^M \frac{1}{\sqrt{\epsilon}} d\epsilon$$

$$\lesssim \sqrt{MKdRD/\sqrt{n}}$$

$$\lesssim RD\sqrt{\frac{Kd}{n}}$$

Taking expectations over the set $T$ (keeping in mind that $R$ is a function of $T$), by Lemma 23 we get

$$\mathbb{E}_T \sup_{\theta \in \Theta} \left| \frac{1}{n} \sum_{i=1}^n \varepsilon_i l_\theta(x_i, \beta_i) \right| \lesssim \mathbb{E}_T[R] D \sqrt{\frac{Kd}{n}}$$

$$\lesssim \frac{C\left(K, d, D, \frac{1}{\lambda_{\min}}\right)}{\sqrt{n}}$$

This completes the proof of (28). By Markov's inequality, (28) implies that for every $\epsilon > 0$,

$$\Pr_T\left[\sup_{\theta \in \Theta} \left| \widehat{D_{GSM}}(\theta) - D_{GSM}(\theta) \right| > \epsilon\right] \leq \frac{C\left(K, d, D, \frac{1}{\lambda_{\min}}\right)}{\sqrt{n}\epsilon}$$

Thus, for every $\epsilon > 0$,

$$\lim_{n \to \infty} \Pr_T\left[\sup_{\theta \in \Theta} \left| \widehat{D_{GSM}}(\theta) - D_{GSM}(\theta) \right| > \epsilon\right] = 0$$

Thus,

$$\sup_{\theta \in \Theta} \left| \widehat{D_{GSM}}(\theta) - D_{GSM}(\theta) \right| \xrightarrow{p} 0$$

as we need. $\qquad\square$

## G  POLYNOMIAL SMOOTHNESS BOUND: PROOF OF THEOREM 4

First, we need several easy consequences of the machinery developed in Section A.3, specialized to Gaussians appearing in CTLD.

**Lemma 25.** *For all $k \in \mathbb{N}$, we have:*

$$\max_{\beta,i} \mathbb{E}_{x \sim p(x|\beta,i)} \|\Sigma_\beta^{-1}(x - \mu_i)\|_2^{2k} \leq d^k \lambda_{\min}^{-k}$$

*Proof.*

$$
\begin{aligned}
\mathbb{E}_{x \sim p(x|\beta,i)} \|\Sigma_\beta^{-1}(x - \mu_i)\|_2^{2k} &= \mathbb{E}_{z \sim \mathcal{N}(0,I_d)} \|\Sigma_\beta^{-\frac{1}{2}} z\|_2^{2k} \\
&\leq \mathbb{E}_{z \sim \mathcal{N}(0,I_d)} \|\Sigma_\beta^{-1}\|_{OP}^k \|z\|_2^{2k} \\
&\leq \lambda_{\min}^{-k} \mathbb{E}_{z \sim \mathcal{N}(0,I_d)} \|z\|_2^{2k} \\
&\leq d^k \lambda_{\min}^{-k}
\end{aligned}
$$

where the last inequality follows by Lemma 30. □

Combining this Lemma with Lemmas 7 and 8, we get the following corollary:

**Corollary 2.**

$$\max_{\beta,i} \mathbb{E}_{x \sim p(x|\beta,i)} \left\| \frac{\nabla_{\mu_i}^{k_1} \nabla_x^{k_2} p(x|\beta,i)}{p(x|\beta,i)} \right\|^{2k} \lesssim d^{(k_1+k_2)k} \lambda_{\min}^{-(k_1+k_2)k}$$

$$\max_{\beta,i} \mathbb{E}_{(x,\beta) \sim p(x|\beta,i)} \left\| \frac{\nabla_{\mu_i}^{k_1} \Delta_x^{k_2} p(x|\beta,i)}{p(x|\beta,i)} \right\|^{2k} \lesssim d^{(k_1+3k_2)k} \lambda_{\min}^{-(k_1+3k_2)k}$$

Finally, we will need the following simple technical lemma:

**Lemma 26.** *Let $X$ be a vector-valued random variable with finite $\mathrm{Var}(X)$. Then, we have*

$$\|\mathrm{Var}(X)\|_{OP} \leq 6\mathbb{E}\|X\|_2^2$$

*Proof.* We have

$$
\begin{aligned}
\|\mathrm{Var}(X)\|_{OP} &= \left\| \mathbb{E}\left[ (X - \mathbb{E}[X])(X - \mathbb{E}[X])^\top \right] \right\|_{OP} \\
&\leq \mathbb{E}\|X - \mathbb{E}[X]\|_2^2 \tag{33} \\
&\leq 6\mathbb{E}\|X\|_2^2 \tag{34}
\end{aligned}
$$

where (33) follows from the subadditivity of the spectral norm, (34) follows from the fact that

$$\|x + y\|_2^2 = \|x\|_2^2 + \|y\|_2^2 + 2\langle x,y \rangle \leq 3(\|x\|_2^2 + \|y\|_2^2)$$

for any two vectors $x, y$, as well as the fact that by Jensen's inequality, $\|\mathbb{E}[X]\|_2^2 \leq \mathbb{E}\|X\|_2^2$. □

Given this lemma, it suffices to bound $\mathbb{E}\|(\nabla_\theta \nabla_{x,\beta} \log p_\theta^\top \nabla_{x,\beta} \log p_\theta\|_2^2$ and $\mathbb{E}\|\nabla_\theta \Delta_{x,\beta} \log p_\theta\|_2^2$, which are given by Lemma 27 and Lemma 28, respectively.

**Lemma 27.**

$$\mathbb{E}_{(x,\beta) \sim p(x,\beta)} \left\| \nabla_\theta \nabla_{x,\beta} \log p_\theta^\top \nabla_{x,\beta} \log p_\theta \right\|_2^2 \leq \mathrm{poly}\left( D, d, \frac{1}{\lambda_{\min}} \right)$$

*Proof.* Recall that $\theta = (\mu_1, \mu_2, \ldots, \mu_K)$, where each $\mu_i$ is a $d$-dimensional vector, and we are viewing $\theta$ as a $dK$-dimensional vector.

$$
\mathbb{E}_{(x,\beta)\sim p(x,\beta)} \left\| \nabla_\theta \nabla_{x,\beta} \log p_\theta^\top \nabla_{x,\beta} \log p_\theta \right\|_2^2
$$

$$
\leq \mathbb{E}_{(x,\beta)\sim p(x,\beta)} \left[ \|\nabla_\theta \nabla_{x,\beta} \log p_\theta\|_{OP}^2 \|\nabla_{x,\beta} \log p_\theta\|_2^2 \right]
$$

$$
\leq \sqrt{\mathbb{E}_{(x,\beta)\sim p(x,\beta)}\|\nabla_\theta \nabla_{x,\beta} \log p_\theta\|_{OP}^4} \sqrt{\mathbb{E}_{(x,\beta)\sim p(x,\beta)}\|\nabla_{x,\beta} \log p_\theta\|_2^4}
$$

where the last step follows by Cauchy-Schwartz. To bound both factors above, we will essentially first use Lemma 10 to relate moments over the mixture, with moments over the components of the mixture. Subsequently, we will use estimates for a single Gaussian, i.e. Corollaries 2 and 1.

Proceeding to the first factor, we have:

$$
\mathbb{E}_{(x,\beta)\sim p(x,\beta)} \|\nabla_{x,\beta}\nabla_\theta \log p_\theta(x,\beta)\|_{OP}^4
$$

$$
\lesssim \mathbb{E}_{(x,\beta)\sim p(x,\beta)} \|\nabla_x \nabla_\theta \log p_\theta(x,\beta)\|_{OP}^4 + \mathbb{E}_{(x,\beta)\sim p(x,\beta)} \|\nabla_\beta \nabla_\theta \log p_\theta(x,\beta)\|_2^4
$$

$$
\lesssim \mathbb{E}_{(x,\beta)\sim p(x,\beta)} \|\nabla_x \nabla_\theta \log p_\theta(x|\beta)\|_{OP}^4 + \mathbb{E}_{(x,\beta)\sim p(x,\beta)} \|\nabla_\beta \nabla_\theta \log p_\theta(x|\beta)\|_2^4
$$

$$
\lesssim \max_{\beta,i} \mathbb{E}_{x\sim p(x|\beta,i)} \|\nabla_x \nabla_\theta \log p_\theta(x|\beta,i)\|_{OP}^4 + \max_{\beta,i} \mathbb{E}_{x\sim p(x|\beta,i)} \|\nabla_\beta \nabla_\theta \log p_\theta(x|\beta,i)\|_2^4 \quad (35)
$$

$$
\leq \text{poly}(d, 1/\lambda_{\min}) \quad (36)
$$

where (35) follows from Lemma 10, and (36) follows by combining Corollaries 2 and 1 and Lemma 15.

The second factor is handled similarly[4]. We have:

$$
\mathbb{E}_{(x,\beta)\sim p(x,\beta)}\|\nabla_{x,\beta} \log p_\theta(x,\beta)\|_2^4
$$

$$
\lesssim \mathbb{E}_{(x,\beta)\sim p(x,\beta)}\|\nabla_x \log p_\theta(x,\beta)\|_2^4 + \mathbb{E}_{(x,\beta)\sim p(x,\beta)} (\nabla_\beta \log p_\theta(x,\beta))^4
$$

$$
\lesssim \mathbb{E}_{(x,\beta)\sim p(x,\beta)}\|\nabla_x \log p_\theta(x|\beta)\|_2^4 + \mathbb{E}_{(x,\beta)\sim p(x,\beta)} (\nabla_\beta \log p_\theta(x|\beta))^4 + \mathbb{E}_{\beta\sim r(\beta)} (\nabla_\beta \log r(\beta))^4
$$

$$
\lesssim \max_{\beta,i} \mathbb{E}_{x\sim p(x|\beta,i)}\|\nabla_x \log p_\theta(x|\beta,i)\|_2^4 + \max_{\beta,i} \mathbb{E}_{x\sim p(x|\beta,i)}(\nabla_\beta \log p_\theta(x|\beta,i))^4 + \max_\beta (\nabla_\beta \log r(\beta))^4
$$

$$
\quad (37)
$$

$$
\leq \text{poly}(d, D, 1/\lambda_{\min}) \quad (38)
$$

where (37) follows from Lemma 10, and (38) follows by combining Corollaries 2 and 1 and Lemma 15, as well as the fact that $\max_\beta (\nabla_\beta \log r(\beta))^4 \lesssim D^8 \lambda_{\min}^{-4}$ by a direct calculation.

Together the estimates (36) and (38) complete the proof of the lemma. $\square$

**Lemma 28.**

$$
\mathbb{E}_{(x,\beta)\sim p(x,\beta)}\|\nabla_\theta \Delta_{x,\beta} \log p_\theta(x,\beta)\|_2^2 \leq \text{poly}\left(d, \frac{1}{\lambda_{\min}}\right)
$$

*Proof.*

$$
\nabla_\theta \Delta_{(x,\beta)} \log p_\theta(x,\beta) \quad (39)
$$

$$
= \nabla_\theta \Delta_x \log p_\theta(x,\beta) + \nabla_\theta \nabla_\beta^2 \log p_\theta(x,\beta) \quad (40)
$$

$$
= \nabla_\theta \Delta_x \log p_\theta(x|\beta) + \nabla_\theta \Delta_x \log r(\beta) + \nabla_\theta \nabla_\beta^2 \log p_\theta(x|\beta) + \nabla_\theta \nabla_\beta^2 \log r(\beta)
$$

$$
= \nabla_\theta \Delta_x \log p_\theta(x|\beta) + \nabla_\theta \nabla_\beta^2 \log p_\theta(x|\beta) \quad (41)
$$

where (39) follows by exchanging the order of derivatives, (40) since $\beta$ is a scalar, so the Laplacian just equals to the Hessian, (41) by dropping the derivatives that are zero in the prior expression.

---

[4]Note, $\nabla_\beta f(\beta)$ for $f : \mathbb{R} \to \mathbb{R}$ is a scalar, since $\beta$ is scalar.

To bound both summands above, we will essentially first use Lemma 10 to relate moments over the mixture, with moments over the components of the mixture. Subsequently, we will use estimates for a single Gaussian, i.e. Corollaries 1 and 2. Precisely, we have:

$$
\begin{aligned}
&\mathbb{E}_{(x,\beta)\sim p(x,\beta)}\|\nabla_\theta \Delta_{x,\beta}\log p_\theta\|_2^2 \\
&\lesssim \mathbb{E}_{(x,\beta)\sim p(x,\beta)}\|\nabla_\theta \operatorname{Tr}(\nabla_x^2 \log p_\theta(x|\beta))\|_2^2 + \mathbb{E}_{(x,\beta)\sim p(x,\beta)}\|\nabla_\theta \nabla_\beta^2 \log p_\theta(x|\beta)\|_2^2 \\
&\lesssim \max_{\beta,i}\mathbb{E}_{x\sim p(x|\beta,i)}\left\|\frac{\nabla_\theta \Delta_x p_\theta(x|\beta,i)}{p_\theta(x|\beta,i)}\right\|_2^2 + \max_{\beta,i}\mathbb{E}_{x\sim p(x|\beta,i)}\left\|\frac{\nabla_\theta \nabla_x p_\theta(x|\beta,i)}{p_\theta(x|\beta,i)}\right\|_{OP}^4 \qquad (42)\\
&\leq \operatorname{poly}(d, 1/\lambda_{\min}) \qquad\qquad (43)
\end{aligned}
$$

where (42) follows from Lemma 10 and Lemma 15, and (43) follows by combining Corollaries 1 and 2.

$\square$

## H  TECHNICAL LEMMAS

### H.1  MOMENTS OF A CHI-SQUARED RANDOM VARIABLE

For the lemmas in this subsection, we consider a random variable $z \sim \mathcal{N}(0, I_d)$ and random variable $x \sim \mathcal{N}(\mu, \Sigma)$ where $\|\mu\| \leq D$ and $\Sigma \preceq \sigma_{\max}^2 I$.

**Lemma 29** (Norm of Gaussian). *The random variable $z$ enjoys the bound*

$$\mathbb{E}\|z\|_2 \leq \sqrt{d}$$

*Proof.*

$$
\begin{aligned}
\left(\mathbb{E}\|z\|_2\right)^2 &\leq \mathbb{E}\|z\|_2^2 \qquad\qquad (44)\\
&= \mathbb{E}\sum_{i=1}^d z_i^2 \\
&= d \qquad\qquad\qquad (45)
\end{aligned}
$$

where (44) follows from Jensen, and (45) by plugging in the mean of a chi-squared distribution with $d$ degree of freedom. $\square$

**Lemma 30** (Moments of Gaussian). *Let $z \sim \mathcal{N}(0, I_d)$. For $l \in \mathbb{Z}^+$, $\mathbb{E}\|z\|_2^{2l} \lesssim d^l$.*

*Proof.* The key observation required is $\|z\|_2^2 = \sum_{i=1}^d z_i^2$ is a Chi-Squared distribution of degree $d$.

$$
\begin{aligned}
\mathbb{E}\|z\|_2^{2l} = \mathbb{E}\left(\|z\|_2^2\right)^l = \mathbb{E}_{q\sim\chi^2(d)}q^l \\
= \frac{(d+2l-2)!!}{(d-2)!!} \leq (d+2l-2)^l \\
\lesssim d^l
\end{aligned}
$$

$\square$

## I  ADDITIONAL RELATED WORK

**Decomposition theorems and mixing times**  The mixing time bounds we prove for CTLD rely on decomposition techniques. At the level of the state space of a Markov Chain, these techniques "decompose" the Markov chain by partitioning the state space into sets, such that: (1) the mixing time of the Markov chain inside the sets is good; (2) the "projected" chain, which transitions between sets with probability equal to the probability flow between sets, also mixes fast. These techniques also can be thought of through the lens of functional inequalities, like Poincaré and Log-Sobolev inequalities.

Namely, these inequalities relate the variance or entropy of functions to the Dirichlet energy of the Markov Chain: the decomposition can be thought of as decomposing the variance/entropy inside the sets of the partition, as well as between the sets.

Most related to our work are Ge et al. (2018); Moitra & Risteski (2020); Madras & Randall (2002), who largely focus on decomposition techniques for bounding the Poincaré constant. Related "multi-scale" techniques for bounding the log-Sobolev constant have also appeared in the literature Otto & Reznikoff (2007); Lelièvre (2009); Grunewald et al. (2009).

**Learning mixtures of Gaussians**   Even though not the focus of our work, the annealed score-matching estimator with the natural parametrization (i.e. the unknown means) can be used to learn the parameters of a mixture from data. This is a rich line of work with a long history. Identifiability of the parameters from data has been known since the works of Teicher (1963); Yakowitz & Spragins (1968). Early work in the theoretical computer science community provided guarantees for clustering-based algorithms (Dasgupta, 1999; Sanjeev & Kannan, 2001); subsequent work provided polynomial-time algorithms down to the information theoretic threshold for identifiability based on the method of moments (Moitra & Valiant, 2010; Belkin & Sinha, 2010); even more recent work tackles robust algorithms for learning mixtures in the presence of outliers (Hopkins & Li, 2018; Bakshi et al., 2022); finally, there has been a lot of interest in understanding the success and failure modes of practical heuristics like expectation-maximization (Balakrishnan et al., 2017; Daskalakis et al., 2017).

**Techniques to speed up mixing time of Markov chains**   SDEs with different choices of the drift and covariance term are common when designing faster mixing Markov chains. A lot of such schemas "precondition" by a judiciously chosen $D(x)$ in the formalism of equation (7). A particularly common choice is a Newton-like method, which amounts to preconditioning by the Fisher matrix (Girolami & Calderhead, 2011; Li et al., 2016; Simsekli et al., 2016), or some cheaper approximation thereof. More generally, non-reversible SDEs by judicious choice of $D, Q$ have been shown to be quite helpful practically (Ma et al., 2015)

"Lifting" the Markov chain by introducing new variables is also a very rich and useful paradigms. There are many related techniques for constructing Markov Chains by introducing an annealing parameter (typically called a "temperature"). Our chain is augmented by a temperature random variable, akin to the simulated tempering chain proposed by Marinari & Parisi (1992). In parallel tempering (Swendsen & Wang, 1986; Hukushima & Nemoto, 1996), one maintains multiple particles (replicas), each evolving according to the Markov Chain at some particular temperature, along with allowing swapping moves. Sequential Monte Carlo (Yang & Dunson, 2013) is a related technique available when gradients of the log-likelihood can be evaluated.

Analyses of such techniques are few and far between. Most related to our work, Ge et al. (2018) analyze a variant of simulated tempering when the data distribution looks like a mixture of (unknown) Gaussians with identical covariance, and can be accessed via gradients to the log-pdf. We compare in more detail to this work in Section 4. In the discrete case (i.e. for Ising models), Woodard et al. (2009b;a) provide some cases in which simulated and parallel tempering provide some benefits to mixing time.

Another way to "lift" the Markov chain is to introduce a velocity variable, and come up with "momentum-like" variants of Langevin. The two most widely known ones are underdamped Langevin and Hamiltonian Monte Carlo. There are many recent results showing (both theoretically and practically) the benefit of such variants of Langevin, e.g. (Chen & Vempala, 2019; Cao et al., 2023). The proofs of convergence times of these chains is unfortunately more involved than merely a bound on a Poincaré constant (in fact, one can prove that they don't satisfy a Poincaré constant) — and it's not so clear how to "translate" them into a statistical complexity analysis using the toolkit we provide in this paper. This is fertile ground for future work, as score losses including a velocity term have already shown useful in training score-based models (Dockhorn et al., 2021).

