# OpenReview forum: "Fit Like You Sample: Sample-Efficient Generalized Score Matching from Fast Mixing Diffusions"
_ICLR.cc/2024/Conference — Submitted to ICLR 2024_

### Official Review · Reviewer_BbBg · 2023-10-29

**Soundness:** 3 good
**Presentation:** 2 fair
**Contribution:** 2 fair
**Rating:** 5
**Confidence:** 2

**Summary:**

The paper studies the problem of using score matching to learn the probability distribution in energy-based models.
In energy-based models, when we learn the probability distribution, we often encounter an intractable normalizing factor.
To avoid this intractable factor, one can use score matching instead.
However, score matching can be statistically less efficient.
This paper works on the connection between the mixing time of a broad class of continuous, time-homogeneous Markov processes with stationary distribution and generator, and the statistical efficiency of an appropriately chosen generalized score matching loss.

**Strengths:**

- The problem seems well-motivated.

**Weaknesses:**

- The presentation is fairly technical and may pose challenges for readers who are not an expert in this particular area.

**Questions:**

Note:
- Theorem 2: What are $\Gamma_{SM}$ and $\Gamma_{MLE}$?

---

> ### Author Response · Authors · 2023-11-14
> **Thank you for your feedback**
>
> Thank you for your feedback! We are glad you find the problem well-motivated. We are happy to make changes to the presentation if you have concrete suggestions on how to make it more approachable.
>
> Regarding your question: $\Gamma_{SM}$ and $\Gamma_{MLE}$ are the covariance matrices of the asymptotic limit for the score matching and maximum likelihood estimators respectively (i.e. the matrix $\Gamma$, s.t. for the estimator $\hat{\theta}_n$, we have $\sqrt{n}(\hat{\theta}_n - \theta^*) \to \mathcal{N}(0, \Gamma)$).

---

> ### Author Response · Authors · 2023-11-21
> **Any outstanding concerns?**
>
> Dear reviewer BbBg,
>
> thank you again for your feedback !
>
> We'd like to remind you that the discussion period is ending tomorrow (11/22), and wanted to follow up and check if there are any outstanding concerns or questions we could answer. If our response (as well as the response to the other reviewers) answered your questions, we hope you'll consider raising your score.
>
> Best,
>
> --Authors

---

> > ### Comment · Reviewer_BbBg · 2023-11-22
> >
> > Thanks for the response. I will take it into consideration during the AC discussion.

---

### Official Review · Reviewer_ufaE · 2023-11-01

**Soundness:** 4 excellent
**Presentation:** 2 fair
**Contribution:** 2 fair
**Rating:** 5
**Confidence:** 2

**Summary:**

The authors study generalized score matching loss, that uses an arbitrary linear operator instead of $\nabla_x$ in the standard score matching objective. They generalize the result of  Koehler et al. (2022) to this setting. Concretely, they show that for a Markov process with stationary distribution $p$ proportional to $\exp(-f(x))$ and a generator $\mathcal{L}$ with Poincare constant $C_P$, one can choose a linear operator such that the error of the corresponding generalized score matching estimator (more precisely, the spectral norm of the covariance of limit distribution, assuming asymptotic normality) can be bounded in terms of $C_P$ and the error of MLE (more precisely, the spectral norm of the covariance of the limit distribution). In addition, they use generalized score matching with additional temperature variable for learning means of Gaussian mixtures with the same covariance matrix.

**Strengths:**

The results are new and non-trivial. The statements are clear, as well as comparison with results from prior works on score matching.

**Weaknesses:**

There are a few things that concern me. So far I'm not convinced that the paper is above the acceptance threshold. Please see the questions below.

**Questions:**

Regarding Theorem 2: Could you please explain if there is any interesting technical contribution compared to Theorem 2 in Koehler et al. (2022)? The proof looks like a straightforward generalization of their proof to your settings, or did I miss anything important?

Regarding Theorem 5:

1) The estimator that you use here doesn't seem to be efficiently computable, is that correct? The score matching estimator for exponential families from Koehler et al. (2022) is efficiently computable (please correct me if I'm wrong), so their motivation to study it and compare with MLE is clear to me. What is the motivation of usage of your estimator for this problem if it is not efficiently computable?

2) As I understood, you are interested in the regime $K \gg d$, so the fact that $C$ from your bound $\Vert \Gamma_{SM} \Vert_{OP} \le C \Vert \Gamma_{MLE}\Vert_{OP}^2$ does not depend on $K$ is important and nice. However, as you said in the footnote, $ \Vert \Gamma_{MLE}\Vert_{OP}$ may depend on $K$, so $K$ can appear in the end in the error, i.e. $\Vert \hat{\mu_i} - \mu^*_i \Vert$
may depend on $K$ even when the corresponding error for MLE doesn't. So it is not clear to me why this dependence on $K$ was important from the very beginning.

If it was really important, then it would make sense to bound not
$\Vert \Gamma_{SM} \Vert_{OP}$, but the largest diagonal entry of $\Gamma_{SM}$ in terms of the largest diagonal entry of $\Gamma_{MLE}$. In this case , if there is such a bound with a factor that does not depend on $K$, then it should imply a bound on $\Vert \hat{\mu_i} - \mu^*_i \Vert$ that does not depend on $K$ (as long as corresponding errors of MLE do not depend on $K$). Is it possible to derive such a bound?

3) There is no comparison with prior works on Gaussian mixtures. While you refer to some of these works in the paper, it is not immediately clear how your result is comparable with them. I think it makes sense to add such a comparison.

4) Can your approach be generalized to more general mixtures of Gaussians, when not all of them have the same covariance (but, say, when all covariances have condition number bounded by $O(1)$)?

And a minor thing:

In Definition 1, is it really fine to use linear operators between the spaces of *all* functions? E.g. in Lemma 1 you use adjoint operators and assume that the operators are between Hilbert spaces.

---

> ### Author Response · Authors · 2023-11-13
> **Thank you for your feedback!**
>
> Thank you for your feedback and questions! We are glad you found our results new and non-trivial, and our writing clear. Hopefully the clarifications below ameliorate your concerns.
>
> **Contribution of Theorem 2**: You are correct that our results are a generalization of Koehler et al. (2022). While the overall proof strategy is similar, Theorem 2 is important in two ways:
>
> (1) **Conceptually** our result makes it clear that there is nothing special about *Langevin diffusion* (and its mixing time, as captured through the standard Poincare constant). In particular, for *any diffusion* of the form (7), one can connect the statistical complexity of the corresponding generalized score matching loss to the mixing time of the diffusion. This provides a “dictionary” to translate between fast-mixing diffusions and corresponding score losses with good statistical complexity. Furthermore, the results in Koehler et al. (2022) also only apply to exponential families—our proof shows this restriction is not needed.
>
> (2) **Technically** the main new ingredient is Lemma 4 (along with the calculations in Lemmas 3, 11 and Lemma 13) which connects the Hessian of the generalized score matching loss to the Dirichlet form of the corresponding diffusion. Again, these calculations are completely new compared to Koehler et al. (2022).
>
>
> **Clarification on Theorem 5**:
>
> *(Q1) Computational efficiency of training score loss, motivation*: The general motivation behind score-matching methods is to avoid calculating a partition function, by fitting the score function through the integration by parts formula (eq (2)). In our case, as in Proposition 6, the CTLD score loss can also be rewritten by integration by parts as an expectation of expressions involving the fitted scores.
>
> Note that even for basic score matching, the corresponding loss may not be convex, so all that’s guaranteed is that a gradient-based method can be efficiently run (which is not the case for maximum likelihood due to the partition function). Nevertheless, there **is** an algorithmic gain compared to maximum likelihood (the possibility of running a gradient-based algorithm efficiently)—so the question of understanding the "statistical cost” (how much worse it is compared to maximum likelihood) is meaningful.
>
> *(Q2) Dependence on K*: the reason why we phrase the result as $\lVert\Gamma_{SM}\rVert_{OP} \leq C \lVert\Gamma_{MLE}\rVert^2_{OP}$ is that we are trying to compare how much (statistically) less efficient score matching is compared to MLE. We stress the point about $C$ not depending on $K$ for two reasons:
>
> (1) While MLE could potentially depend on $K$—the dependence is likely to be mild (we are estimating $K$ vectors, so the dependence would be no worse than linear); if $C$ depends poorly (e.g. exponentially in $K$) it would render our result uninteresting for mixtures with large number of components. Moreover, since MLE is asymptotically statistically optimal, whatever dependence on $K$ it incurs is unavoidable.
>
> (2) Theorem 2 proceeds via a decomposition result, similar as prior analyses of simulated tempering (Lee et al, Ge et al (2018)) — and all these results incur a dependence on $K$. This is because their setup is somewhat different — in particular, they analyze discrete tempering, where the annealing is done via temperature scaling (rather than Gaussian convolution).
>
> *(Q3) Comparison with prior work on Gaussian mixture models*: We are happy to expand on the discussion in Appendix I. Mostly though, these results are incomparable: they focus on end-to-end provable algorithms for learning mixtures in various regimes—typically trying to deal with the case of mixtures with not-well-separated means. Our focus is on understanding the *statistical complexity of score matching losses*—not on providing new end-to-end provable algorithms for Gaussian mixtures. (Note, the CTLD loss is not convex, and we don’t analyze what gradient descent or related algorithms would converge to.) Gaussian mixtures are just chosen as a very natural toy model for multimodal data, and as a “proof of concept” that annealed score matching is statistically well-behaved for many distributions for which vanilla score matching is not (via the lower bound in Koehler et al (2022)).
>
> *(Q4) Gaussians with different covariance*: it is quite likely that our results apply to mixtures with covariances which are $1+O(1/d)$ away from each other, but not larger (were d is the ambient dimension). The reason for this is that annealing “distorts” exponentially the relative volumes of the Gausians (technically, what breaks is the chi-squared divergence bound in Lemma 21). However, we note that for universal approximation using mixtures, one can use a mixture of (exponentially many) Gaussians with covariance $\epsilon I$ to $\epsilon$-approximate the distribution of choice.

---

> ### Author Response · Authors · 2023-11-21
> **Any outstanding concerns?**
>
> Dear reviewer ufaE,
>
> thank you again for your feedback and questions !
>
> We'd like to remind you that the discussion period is ending tomorrow (11/22), and wanted to follow up and check if there are any outstanding concerns or questions we could answer. If our response clarified your original concerns, since you otherwise thought our results are clear, new and non-trivial, we hope you'll consider raising your score.
>
> Best,
>
> --Authors

---

> > ### Comment · Reviewer_ufaE · 2023-11-23
> >
> > Dear Authors,
> >
> > Thank you very much for the clarification! I don't have further questions. Currently I don't increase the score, but I will take into account your response during the Reviewer/AC Discussion.

---

### Official Review · Reviewer_v9Ko · 2023-11-06

**Soundness:** 3 good
**Presentation:** 3 good
**Contribution:** 3 good
**Rating:** 8
**Confidence:** 3

**Summary:**

This paper proposes a general framework for designing generalized score matching losses with good sample complexity from fast-mixing diffusions. More precisely, for a broad class of diffusions with generator $\mathcal{L}$ and Pincare constant $C_P$, they can choose a linear operator $\mathcal{O}$ such that  the generalized score matching loss $E[\|\mathcal{O} p / p - \mathcal{O} p_{\theta} / p_{\theta}\|_2^2] / 2$ has a statistical complexity that is a factor $C_P^2$ worse than that of maximum likelihood. In addition, they analyze a lifted diffusion, which introduces a new variable for temparature and provably show statistical benefits of annealing for score matching. They apply their approach to sample from Gaussian mixture distributions.Their first result generalizes that of Koehler 2022.

**Strengths:**

The paper is well motivated and well written. It generalizes a previous paper on score matching (Koehler 2022) to generalized linear operator and correspondingly general score matching loss. The authors are also able to design a Markov chain termed CTLD based on the idea of anneling. Motivated by this chain, they are able to estimate the score function for Gaussian mixture distribution that has multiple modes and control the generalized score matching loss. The framework they propose is novel and quite interesting.

**Weaknesses:**

Several assumptions in the paper seem abit strong, and it would be good if the authors can elaborate a bit more on them. For the GMM application, it would be good to compare their result with the previous ones. Finally, I would love to see an experiment that supports their result, but the result itself is also interesting enough.

**Questions:**

1. Assumption 1 and 2 of Theorem 2 seems pretty strong. Could the authors give an example where these assumptions hold when $\mathcal{O} \neq \nabla_x$ and not from CTLD? In general, how do we validate these assumptions?

2. What is $\mathcal{O}$ for CTLD?

3. Maybe the authors can comment a bit on how Theorem 5 compares with the previous results, in particular, can the results of Koehler 2022 also be applied to get an upper bound?

4. Can we apply generalized score matching loss to diffusion sampling? Maybe the authors can comment a bit on the feasiblity of that.

---

> ### Author Response · Authors · 2023-11-14
> **Thank you for the encouraging feedback!**
>
> Thank you for your encouraging review! We are glad you found our framework novel and interesting, and our paper well-written.
>
> *(Q1) Strength of Assumptions 1 and 2*: Assumption 1 is meant to capture a broad class of multimodal distributions. Moreover, by the universal approximation property of mixtures, any distribution can be approximated by a mixture of sufficiently many Gaussians. Assumption 2 is meant to check that for the “canonical” parametrization of a mixture, the smoothness term in the asymptotic sample complexity (Theorem 4) can be effectively bounded. Some kind of weight-tieing in the parametrization of the score at different temperatures has to be assumed to bound the smoothness term (else, the scores the model estimates at different temperatures need not have anything to do with each other!) It would be a great direction for future research to understand what assumption suffices to have a good bound on this term (we certainly don't think Assumption 2 is necessary). Our goal was to check that the “canonical” parametrization results in a good bound.
>
> *(Q2) What is O for CTLD*: $\mathcal{O}$ is $\nabla_x$ for CTLD (the Markov Chain is however over the augmented space that includes the temperature random variable).
>
> *(Q3) Theorem 5, relation to Koehler et al (2022)*: The results in Koehler et al (2022) are for the basic version of score matching (eq (2)), and they incur a bound that depends on the Poincare constant of the distribution, which can be quite bad for well-separated mixtures. For instance, for a mixture of even two Gaussians with means $-\mu$ and $\mu$ and variance $\sigma^2$, the Poincare constant scales exponentially in $\mu^2/\sigma^2$ [1]. By contrast, our results scale polynomially in the norms of the means, the ambient dimension, and minimum and maximum eigenvalues of the covariance matrices.
>
> *(Q4) Using CTLD for diffusion models*: The CTLD loss can plausibly be used for better diffusion model training: the primary obstacle is the fact that it involves derivatives of the score, which is a $d \times d$ matrix. As we note after Proposition 3, similar “high-order” analogues of score losses have been proposed in some works (e.g. [2]), with some success. Some preconditioning strategies for sampling from diffusion models have also been recently proposed [3], again with some success.
>
> [1] Chafai, Djalil, and Florent Malrieu. "On fine properties of mixtures with respect to concentration of measure and Sobolev type inequalities." In Annales de l'IHP Probabilités et statistiques, vol. 46, no. 1, pp. 72-96. 2010.
>
> [2] Meng, Chenlin, Yang Song, Wenzhe Li, and Stefano Ermon. "Estimating high order gradients of the data distribution by denoising." Advances in Neural Information Processing Systems 34 (2021): 25359-25369.
>
> [3] Zhang, Qinsheng, and Yongxin Chen. "Fast Sampling of Diffusion Models with Exponential Integrator." In The Eleventh International Conference on Learning Representations. 2022.

---

### Meta-Review · Area_Chair_FEiQ · 2023-12-20

**Metareview:**

The paper presents theoretical results connecting the mixing time of a broad class of Markov processes to generalized score matching. This allows the authors to leverage methods to improve the mixing time of Markov chains (e.g. simulated tempering)  to construct score-matching losses with a provable better sample efficiency.

The theoretical results are interesting and to the best of my knowledge novel and thus form an interesting contribution. The paper, however, would benefit from an improved presentation (currently a conclusion is missing and several formulas are too, correcting this would probably make the paper longer than 9 pages, and adding intuitive descriptions would help readers less familiar with the topic) and from a simulation study demonstrating the benefits of the sample-efficient score matching loss. Without the latter, it is not completely clear, if the novel theoretical results are of practical interest.

**Justification For Why Not Higher Score:**

Representation could be improved (which has not been done during rebuttal period) and some simulation study is missing.

**Justification For Why Not Lower Score:**

N/A

---

### Decision · Program_Chairs · 2024-01-16

Reject